# Chemical Compounds of Berry-Derived Polyphenols and Their Effects on Gut Microbiota, Inflammation, and Cancer

**DOI:** 10.3390/molecules27103286

**Published:** 2022-05-20

**Authors:** Abdelhakim Bouyahya, Nasreddine El Omari, Naoufal EL Hachlafi, Meryem El Jemly, Maryam Hakkour, Abdelaali Balahbib, Naoual El Menyiy, Saad Bakrim, Hanae Naceiri Mrabti, Aya Khouchlaa, Mohamad Fawzi Mahomoodally, Michelina Catauro, Domenico Montesano, Gokhan Zengin

**Affiliations:** 1Laboratory of Human Pathologies Biology, Department of Biology, Faculty of Sciences, Mohammed V University in Rabat, Rabat 10106, Morocco; 2Laboratory of Histology, Embryology, and Cytogenetic, Faculty of Medicine and Pharmacy, Mohammed V University in Rabat, Rabat 10100, Morocco; nasrelomari@gmail.com; 3Microbial Biotechnology and Bioactive Molecules Laboratory, Sciences and Technologies Faculty, Sidi Mohmed Ben Abdellah University, Imouzzer Road Fez, Fez 30003, Morocco; naoufal.elhachlafi@usmba.ac.ma; 4Faculty of Pharmacy, University Mohammed VI for Health Science, Casablanca 82403, Morocco; eljemli.meryem@gmail.com; 5Laboratory of Biodiversity, Ecology, and Genome, Faculty of Sciences, Mohammed V University in Rabat, Rabat 10106, Morocco; maryam.hakkour@gmail.com (M.H.); balahbib.abdo@gmail.com (A.B.); 6Laboratory of Pharmacology, National Agency of Medicinal and Aromatic Plants, Taounate 34025, Morocco; nawal.elmenyiy@usmba.ac.ma; 7Molecular Engineering, Valorization and Environment Team, Polydisciplinary Faculty of Taroudant, Ibn Zohr University, Agadir 80000, Morocco; s.bakrim@hotmail.com; 8Laboratory of Pharmacology and Toxicology, Bio Pharmaceutical and Toxicological Analysis Research Team, Faculty of Medicine and Pharmacy, Mohammed V University, Rabat 10000, Morocco; naceiri.mrabti.hanae@gmail.com; 9Laboratory of Biochemistry, National Agency of Medicinal and Aromatic Plants, Taounate 34025, Morocco; aya.khouchlaa@gmail.com; 10Department of Health Sciences, Faculty of Medicine and Health Sciences, University of Mauritius, Reduit 80837, Mauritius; f.mahomoodally@uom.ac.mu; 11Department of Engineering, University of Campania “Luigi Vanvitelli”, Via Roma 29, 81031 Aversa, Italy; 12Department of Pharmacy, University of Naples Federico II, Via D. Montesano 49, 80131 Naples, Italy; domenico.montesano@unina.it; 13Physiology and Biochemistry Research Laboratory, Department of Biology, Science Faculty, Selcuk University, 42130 Konya, Turkey

**Keywords:** berries, polyphenols, gut microbiota, cancer, metabolic disorders

## Abstract

Berry-derived polyphenols are bioactive compounds synthesized and secreted by several berry fruits. These polyphenols feature a diversity of chemical compounds, including phenolic acids and flavonoids. Here, we report the beneficial health effects of berry-derived polyphenols and their therapeutical application on gut-microbiota-related diseases, including inflammation and cancer. Pharmacokinetic investigations have confirmed the absorption, availability, and metabolism of berry-derived polyphenols. In vitro and in vivo tests, as well as clinical trials, showed that berry-derived polyphenols can positively modulate the gut microbiota, inhibiting inflammation and cancer development. Indeed, these compounds inhibit the growth of pathogenic bacteria and also promote beneficial bacteria. Moreover, berry-derived polyphenols exhibit therapeutic effects against different gut-microbiota-related disorders such as inflammation, cancer, and metabolic disorders. Moreover, these polyphenols can manage the inflammation via various mechanisms, in particular the inhibition of the transcriptional factor Nf-κB. Berry-derived polyphenols have also shown remarkable effects on different types of cancer, including colorectal, breast, esophageal, and prostate cancer. Moreover, certain metabolic disorders such as diabetes and atherosclerosis were also managed by berry-derived polyphenols through different mechanisms. These data showed that polyphenols from berries are a promising source of bioactive compounds capable of modulating the intestinal microbiota, and therefore managing cancer and associated metabolic diseases. However, further investigations should be carried out to determine the mechanisms of action of berry-derived polyphenol bioactive compounds to validate their safety and examinate their clinical uses.

## 1. Introduction

The worldwide dietary guidelines recommend a daily consumption of fruits and vegetables due to their contributions in providing vitamins, minerals, and phytochemicals to the body [1]. Among fruits, berries are of particular importance due to their generally higher content of bioactive compounds [2]. Botanically, berry fruits represent a variety of small fleshy fruits produced from a single ovary, characterized by red, purple, and blue colors [3]. The most common are: strawberry, currant, blueberry, white and black mulberry, rosehip, raspberry, blackberry, barberry, and gooseberry [4]. They are widely distributed in Asia, Europe, all Anatolian and Mediterranean countries, and North African countries [5]. These fruits are generally consumed not only in fresh and frozen form, but also as processed and derived products, including yogurts, drinks, jams, dried and canned fruits, and jellies [6].

Recently, interest in berries has intensified worldwide due to the potentially useful phytochemicals found in these fruits that contribute to their biological properties [2]. They contain high levels of polyphenols, including flavonoids (anthocyanins, flavonols, and flavanols), condensed and hydrolyzable tannins, phenolic acids, stilbenes, and lignans [7]. Their concentrations vary according to species, genotype, environmental conditions, degree of ripeness, cultivar, cultivation site, processing, and storage of the fruit [8]. All these substances exert a synergistic and cumulative effect on health promotion and prevention of various diseases [9]. In recent years, several clinical and experimental studies have focused on the biological properties of berries and their components. In particular, increasing attention has been paid to the role of berry polyphenols in the modulation of gut microbiota [10] and the prevention of various diseases and disorders such as inflammation [11], cancer [12], and diabetes [13].

This review aims to provide an overview of the main bioactive compounds in berry fruits and their potential modulation of the gut microbiota and gut health, as well as their health-promoting properties in terms of metabolic disorders and various types of cancer. Moreover, data collected from studies on bioavailability, metabolism, and mechanisms of dietary polyphenols from a wide range of berries are discussed, along with technological approaches to enhance the medicinal value of these compounds.

## 2. General Information on Berry-Derived Polyphenols 

### 2.1. Chemical Composition 

Berries are a rich source of phytochemicals, especially phenolic compounds. These compounds are involved in various health effects attributed to berry fruits, such as antioxidant, anti-inflammatory, antimicrobial, anticancer, and antidiabetic activities [1,6,14] They are also implicated in plant growth and protection against biotic and abiotic stresses [15]. The distribution of these polyphenols varies noticeably depending on several factors, including plant genotype, environmental conditions, growing location, and fruit processing and storage [8,16,17].

Berries are known to be a rich source of flavonoid compounds, in particular anthocyanins (up to 5000 mg/kg), which are responsible for the blue, purple, and red colors, as well as the high acceptance of these fruits [18,19]. In addition to their ability to cross the blood–brain barrier, these pigments have potent antioxidant effects and have been widely investigated for their potential bioactivity in the human body [20]. 

Anthocyanins appear to be very instable and susceptible to degradation. In fact, several factors, including oxygen, enzymes, temperature, light, and pH, can affect the chemistry of anthocyanins, and therefore their stability [15,21]. Chemically, anthocyanins are water-soluble glycosides of polyhydroxylated and polymethoxylated derivatives of 2-phenylbenzopyrylium [22]. In general, berry anthocyanins are found as glycosides of their respective aglycones, known as anthocyanidins and derived from delphinidin, cyanidin, pelargonidin, petunidin, malvidin, and peonidin anthocyanidins [23], which form O-linked conjugates with certain sugars, mainly glucose, galactose, rhamnose, arabinose, sambubiose, and rutinose (Figure 1). These molecules are mainly in the form of 3-monosides, 3-biosides, 3-triosides, and 3,5-diglucosides [24,25,26,27]. The profile of berry anthocyanins is particularly diverse and varied from one species to another. Indeed, some berries contain only one type of anthocyanin derivative (cyanidin) such as red currant (*Rubus idaeus*) [8] and black elderberry (*Sambucus nigra* L.) [28], while a wide range of anthocyanins has been reported in blackberry, raspberry, strawberry, and blueberry [25,29]. Pelargonidin 3-glucoside is the major anthocyanin found in strawberries, whereas the 3-*O*-galactosidase and 3-*O*-arabinoside of delphinidin, cyanidin, and malvidin are the main components identified in blueberries *(Vaccinium* spp.) [29,30].

#### 2.1.1. Flavonols and Flavan-3-ols

Flavonols and flavan-3-ols are subgroups of flavonoids found in various berries, including blueberry, strawberry, bilberry, cranberry, lingonberry, blackberry, raspberry, strawberry, chokeberry, and elderberry [26,31,32,33]. Flavonols are mainly represented by quercetin, myricetin, and kaempferol. These compounds have a hydroxyl group at position 3 of the C-ring, which may be substituted with a number of sugars such as rhamnose, galactose, glucose, robinose, and rutinose. Myricetin 3-*O*-rutinoside, myricetin 3-*O*-glucoside, quercetin 3-*O*-glucoside, quercetin-3-arabinoside, and rutin (quercetin 3-*O*-rutinosides) [34,35] are abundantly found in berries. For instance, quercetin 3-galactoside and quercetin 3-glucoside are the most abundant flavonols in blackberry [36].

Quercetin was the predominant flavonol in bog whortleberry (Vaccinium uliginosum) (158 mg/kg) and bilberry (17–30 mg/kg) [8]. In black currant cultivars, myricetin was the main flavonol (89–203 mg/kg), followed by quercetin (70–122 mg/kg) and kaempferol (9–23 mg/kg). Moreover, blackberries contained a flavonol profile complex with the presence of nine quercetin and three kaempferol derivatives, as well as two acylated quercetin-derived compounds [37]. Furthermore, catechin, epicatechin, and epigallocatechin gallate are the most prevalent flavan-3-ols in berries (Figure 1) [31,38]. Sea buckthorn leaves have been found to be the main source of gallocatechin and epicatechin. Flavonols and flavan-3-ols exert strong antioxidant protection and health benefits, and the abundance of these compounds in different berries suggests their contributions as promising cardioprotective, neuroprotective, anti-inflammatory, and anticarcinogenic foods [39].

#### 2.1.2. Ellagitannins

Ellagitannins are hydrolyzable tannins that are widely distributed in berry plants, especially cloudberry (315.1 mg/100 g of fw), red raspberry (297.3 mg/100 g of fw), strawberry (77.1 mg/100 g of fw), and sea buckthorn (1 mg/100 g of fw), in which they are considered the main active compounds [6]. Since ellagitannins occur in various structures, they can be present as monomers, oligomers, or complex polymers. Several ellagitannin monomers, dimers, trimers, and tetramers have been identified in blackberries, but most are found only in seed tissue [40,41,42]. Red raspberries contain a high concentration of ellagitannins (300 mg/kg), mainly in the form of sanguiin H-6 and lambertianin C, which are responsible for most of the health benefits [39,42]. Ellagitanins also were identified in the highest amount in pomegranates (*Punica granatum* L.). The fruits include gallagic acid, an analog of ellagic acid (the major breakdown products of ellagitannins), based on four gallic acid residues, and punicalin, the major predominant monomeric ellagitannin in which l gallagic acid is bound to glucose [42,43]. The high-performance liquid chromatography–electrospray ionization–mass spectrometry and matrix-assisted laser desorption/ionization-time-of-flight mass spectrometry (MALDI-TOF-MS) identified large and diverse ellagitannins in berries, including isomeric forms of pedunculagin, castalagin/vescalagin, galloyl-HHDP glucose, lambertianin C, lambertianin D, galloyl-bis-HHDP glucose, sanguiin H-6/lambertianin A, and sanguiin H-10 [44]. Proanthocyanidins, also called condensed tannins, are dimers, oligomers, and polymers of catechins. Many berries, in particular cranberries, contain proanthocyanidins with an A-type bond [45].

#### 2.1.3. Phenolic Acids

Phenolic acids, including hydroxycinnamates and hydroxybenzoate, have also been reported in berries. These acids are usually found in the conjugated forms of esters and glycosides, and are rarely found as free acids. The main phenolic acids in berries are caffeic, ferulic, ellagic, *p*-coumaric, vanillic, chlorogenic, neochlorogenic, sinapic, and protocatechuic acids, as well as the β-D-glucosides of *p*-coumaric and caftaric acids [38,39,40,41,42,43,44,45,46], while other phenolic acids are present in low concentrations, such as coutaric and caftaric acids [47,48,49] (Figure 2). Berries such as blackberries are a promising source of hydroxybenzoic acids, including p-hydrobenzoic, protocatechuic, gallic, vanillic, salicylic, gentisic acid, and glycosidic and ester forms of salicylic acid [50]. 

Hydroxycinnamic acids such as ferulic, *m*-coumaric, *p*-coumaric, and caffeic acids have been identified in blackberries in free, ester, and glycosidic forms, with a predominance of ester forms of *m*-coumaric, 3,4-dimethoxycinnamic, and hydroxycaffeic acids [36].

### 2.2. Biosynthesis and Regulation 

During their development, grape berries are generally exposed to several biotic and abiotic stimulators, and the regulation (temporal and developmental) of the polyphenol metabolism of these berries responds to various signals/factors. Indeed, exposure of berries to different types of light energies can influence the synthesis of polyphenols and sometimes alter their biochemical characteristics and the expression of regulatory genes [51,52,53,54,55,56,57,58,59,60].

To decipher the genetic mechanisms involved in increasing polyphenol contents (flavonols, anthocyanins, and flavan-3-ols) present in the tissues of grape berries (*Vitis vinifera*), Matus et al. [51] studied the relationship between the expression of different transcriptional regulators, responsible for flavonoid synthesis, and the effect of postveraison solar energy. They revealed that this energy affected the *MYB* genes, particularly those that regulate the later stages of the flavonol or anthocyanin biosynthesis pathway, rather than those that control the different stages of this pathway. 

A subsequent study evaluated the impact of solar ultraviolet-B (UV-B) radiation and abscisic acid (ABA) spray, which have an important role in acclimatization to environmental signals, on the biosynthesis of polyphenols (anthocyanins and nonanthocyanins) in the skin of *V. vinifera* berries [53]. This combination accelerated the accumulation of phenol in the berry skin. In addition, the majority of phenols were increased by UV-B treatment with an additive increase during exogenous application of ABA, with a change in anthocyanin and nonanthocyanin profiles. These findings may explain the use of high-altitude vineyards in winemaking due to their production of berries with a high phenolic compound yield, since they receive high levels of UV-B at these altitudes. This sensitivity of phenolic compounds to light environments was also confirmed a year later by testing the effects of visible light and UV on the biosynthesis of flavonols and proanthocyanidins (PAs) in young berry skins of *V. vinifera* cv. Cabernet Sauvignon [54]. Indeed, visible light and solar UV induced the biosynthesis of PAs and flavonols, respectively, and the exclusion of these types of light showed different decreases in flavonoid biosynthesis. In addition, different regulation of *MYB* transcription factors and structural genes was observed depending on the light environment. 

Moreover, Kondo and colleagues evaluated the effect of red and blue light-emitting diode (LED) irradiation at night on anthocyanin synthesis and their associated genetic expressions, as well as the metabolism of endogenous ABA in grape berries [55]. It was found that blue LED treatment at night increased the concentrations of these natural pigments and the expressions of certain genes (*VvUFGT*, *VlMYBA2*, and *VlMYBA1-2*) in grape skin compared to the red LED, whereas these concentrations did not coincide with those of endogenous ABA, nor with *VvCYP707A1* or *VvNCED1* expressions. Likewise, the regulation of anthocyanin synthesis was determined in another study in teinturier grapes exposed to light [56]. This fruit accumulates anthocyanins in both tissues (skin and flesh) at different concentrations. As a result, the exclusion of light decreased the total amount of these compounds and the transcriptional abundance of structural genes (transporters and regulators) in a tissue- and cultivar (wild-type and teinturier)-specific manner. 

During the same year, the UV-C irradiation/biosynthesis relationship was investigated by a Chinese research team by determining the content of resveratrol, a polyphenol from the stilbenes class very abundant in grapes, in the berries of Beihong (*V. vinifera* × *V. amurensis*) using their gene expressions, and also by distinguishing the stage of development at which this synthesis was more sensitive to this irradiation [57]. This compound has been associated with important anticarcinogenic properties in many ways, and even has chemosensitizing effects (resveratrol chemosensitizes TNF-β-induced survival of 5-FU-treated colorectal cancer cells). 

It has been shown that under natural conditions, the content of resveratrol, before veraison, was very low, whereas it increased (500%) from veraison to maturity. Interestingly, UV-C irradiation strongly stimulated the synthesis of this compound, and more specifically, two weeks before veraison. However, this synthesis gradually decreased, which was explained by the regulation of stilbene synthase, the key enzyme in resveratrol biosynthesis, by the transcription factor *MYB14*. The same authors also revealed that the concentration of this constituent increased in the berry skin of the same species during exogenous application of ABA 10 days before veraison or at veraison [61].

To compare the interactive and independent impacts of these types of solar radiation, ABA application, and water deficit on the berries of *V. vinifera* cv. Malbec, a study was carried out during veraison and two weeks after [58]. In fact, a water deficit is widely used after veraison in regulating the concentration of berry polyphenols. However, in this study, water restriction after veraison did not affect the majority of polyphenols, while exposure to solar UV-B independently increased the total low-molecular-weight phenols (LMWP) in berry skins. In addition, sprayed ABA increased anthocyanins independent of other factors. This was explained by the fact that the increase in polyphenol content was used as a defense mechanism by the tissues against environmental factors (water deficit and UV-B). 

Likewise, a more recent study evaluated the effects of these factors on the phenotype of the same species (*V. vinifera* L. cv. Malbec), and hence the polyphenol content [59]. DNA methylation may regulate gene expression and influence plant growth and acclimation. As already seen in the previous study, UV-B, ABA, and water-deficit treatments increased LMWP in berry skins. Thus, water deficit and UV-B treatments increased the number of DNA methylation changes. This epigenetic mechanism leads to acclimation via the regulation of LMWP accumulation during and after UV-B exposure.

In order to understand the mechanism of hormonal balance under water-deficit irrigation, a previous study was carried out on *V. vinifera* L. berries under different irrigation systems such as partial rootzone drying (PRD) and well-watered [62]. Effectively, the berries subjected to PRD irrigation had a very significant size and yield, and at the beginning of veraison, they presented a higher ABA and free polyamines (phytohormones) than other systems. These results suggest that berry ripening and the subsequent synthesis of polyphenols are strongly linked to the response to the applied water volume, which is likewise linked to hormonal factors. This response was not entirely consistent with that noted by Ojeda and collaborators, who observed a decrease in the berry size of *V. vinifera* cv. Shiraz following the application of various levels of water deficits during berry growth [63]. Despite this, the concentration of phenolic compounds increased in the berry skin. This suggests that the biosynthesis depends on the level of severity and the application period of the water deficit.

Furthermore, Degu and collaborators tested the sensitivity of the detached berries of a Shiraz grapevine to environmental disturbances (stress), in particular temperature (40 °C), light (2500 μmol m^−2^ s^−1^), jasmonic acid (200 μM), ABA (3.026 mM), and oxidative stress, to demonstrate their action on berry phenolic biosynthesis [60]. The authors revealed specific metabolic responses to the signals and stage of berry development. Indeed, they recorded more marked responses to stress in berries at the preveraison stage than in those in veraison.

Along the same lines, Sun et al. [64] analyzed the metabolite and transcriptional profiles of polyphenol biosynthesis in Cabernet Sauvignon grape berries under different solar exposures at various phenological stages. Regarding the accumulation of flavonoids and hydroxycinnamic acids (HCAs), sun exposure has been shown to not alter the concentrations of anthocyanins and flavan-3-ols over seasons, and that it considerably increased those of HCAs and of flavonols. This was well correlated with the expression of genes encoding certain enzymes participating in the biosynthesis of polyphenols, as well as with transcription factors (*MYB*, *WRKY*, *C2H2*, etc.). Genes associated with the biosynthesis and signal transductions of phytohormones such as auxin, ABA, and ethylene have also been identified; indicating the importance of these in polyphenol biosynthesis in grape berries in response to increased exposure to sunlight. 

In contrast, the importance of light in polyphenol biosynthesis was also verified by shading the berries of *V. vinifera* L. cv. Cabernet Sauvignon [65,66]. Indeed, this method allowed the observation of a decrease in PA accumulation in the skins during the development of berries. In addition, a decrease in the transcriptional expression of leucoanthocyanidin reductase (LAR), anthocyanidin reductase (ANR) [65], and *VvF3′5′H* [66] genes, which are involved in polyphenol biosynthesis, have been recorded.

As already mentioned, ABA is a phytohormone that plays a major role in the biosynthesis of polyphenols, and numerous research works have also highlighted this [67,68,69,70,71,72,73,74,75,76,77].

Anthocyanins are the most abundant polyphenols in grape berries. They are harmless red pigments widely used in winemaking that have sparked the interest of several investigations to improve the content of this compound, and therefore the quality of the formulated products. One of the most popular approaches in this regard was the exogenous application of ABA. Indeed, spraying this hormone at different doses on grape organs of different cultivars (Cabernet Sauvignon, Cabernet Franc, Isabel, Noble, and Alachua) significantly increased anthocyanin levels in berries at various development stages [67,69,70,75]. This was in agreement with the results obtained from another hormonal treatment based on the exogenous application of ABA and methyl jasmonate (MeJA) adopted by a Chinese research laboratory in a study of the anthocyanin composition of *V. vinifera* cv. Cabernet Sauvignon berries [73]. 

In fact, MeJA is a volatile organic compound that is naturally synthesized by plants in response to a hazard to produce their defense products, and also to alert nearby plants to the existence of a potential threat. Indeed, the combination of two plant hormones can have promising results in polyphenolic biosynthesis, which was proven by Yang et al. [77] when they combined ABA with melatonin in a spray treatment of *V. vinifera* cv. Kyoho berries. This combination therapy significantly improved flavanol biosynthesis, in particular catechin (200 μM), with a high upregulation of the expression of genes associated with flavonoid biosynthesis. However, ABA and another phytohormone (gibberellin A_3_) independently affected this polyphenolic synthesis in *V. vinifera* cv. Malbec berries [74].

Additionally, ABA not only promotes anthocyanin biosynthesis, but also increases gene expression and transcription factors involved in flavonoid synthesis in grapes [76]. This has solved a problem with many hybrid table grapes grown in the subtropics regarding the deficiency of anthocyanins and pigments in the berries. 

Colorimetric variables can also provide information about the quality of berries and grape juice. In this context, two studies published in the same year by Yamamoto and co-workers on cv. Isabel grape berries showed an increase in the color and total anthocyanin concentration in berries treated with ABA (400 mg/L), and consequently an increase in the color intensity in juices [71,72]. Very recently, this relationship between anthocyanins and *V. vinifera* skin color showed a very strong correlation following the application of ABA at different stages of berry development [68].

On the other hand, the biosynthesis of polyphenols and their accumulation in berries can be distinguished by genetic and epigenetic factors via gene-environment interactions. This was proven by some studies concerning anthocyanin biosynthesis [78,79]. At four stages of ripening, Castellarin and Gaspero, Castellarin and Di Gaspero [78], monitored the regulation of six anthocyanin genes, as well as of four associated transcription factors with evaluation of the correlation between the expression of these genes, anthocyanin concentration, and berry color intensity in nine cultivars of *V. vinifera*. Hence, they found an association between color phenotypes and anthocyanin metabolites with gene expression profiles (transcript-metabolite-phenotype relationship). 

In addition, another *MYB* factor named *VvMYB5b* has been found to have the ability to regulate the biosynthesis of anthocyanin and PA throughout the development of grape berries, providing insight into the transcription mechanisms related to polyphenol pathway regulation [79]. This genetic control was also highlighted by monitoring the accumulation of PAs correlated with the expression of their key genes during the development of “Aglianico” berries [80]. Indeed, the contents of PAs showed differences in distribution depending on the organs and development stages. More importantly, the expression of key genes for these polyphenols was tissue-specific. This explains the capacity to accumulate high polyphenol levels in certain types of vines. The findings of this study corroborated those obtained by Davik et al. [81], who identified the candidate genes controlling the biosynthesis of polyphenolic compounds in strawberries (*Fragaria* × *ananassa* Duch.). 

In another report, PA synthesis began early in the grape development process and ended when ripening began. Additionally, the expression of genes encoding the enzymes ANR and LAR determined the accumulation and composition of PA during the development of *V. vinifera* L. cv. Shiraz berries [82].

On the other hand, the berries of some *V. vinifera* cultivars exhibit unique characteristics compared to other cultivars, including high levels of polyphenolics. To explain this phenomenon, Da Silva and colleagues carried out genome sequencing of the cultivar Tannat, which has remarkable specific properties, for a comparison with the PN40024 reference genome [83]. They discovered the presence of 1873 Tannat-specific genes not shared with PN40024 that contributed to the expression of enzymes involved in polyphenolic biosynthesis and are responsible for the outstanding characteristics of Tannat berries, subsequently explaining the high quality of the products from this cultivar.

In addition to the main factors cited above, there are other factors that also influence the biosynthesis of polyphenols, such as temperature, abiotic stress, application of mineral elements, and ripening disorders.

Effectively, plants use mineral elements for their growth and development. Nitrogen and potassium are among the main elements that a plant needs in large quantities. In fact, the combined nutrition of nitrogen and potassium at different doses can stimulate anthocyanin synthesis in the skin of *Tempranillo* cultivar grapes, thereby affecting the color density of the must during ripening [84]. Likewise, calcium, which is part of the substances used in small quantities by plants, directed the metabolism of “Vinhão” grape berries, through exogenous treatment, toward a high synthesis of phenolic acids, total flavonols, and stilbenoids versus low anthocyanin production [85]. A berry-ripening disorder can also affect this anthocyanin production. This was demonstrated in the shriveled berries of Zweigelt grapes [86]. Generally, the synthesis of metabolites, including polyphenols, is regulated at the transcriptional scale. Indeed, the expression of the genes that control the synthesis of these metabolites is itself controlled by internal and external factors (temperature, stress, soil, pH, etc.). These factors mobilize gene expression by regulating epigenetic control pathways; that is to say, all the factors condition the synthesis of metabolites. This mode of transcriptional regulation is not completely clear, and certainly stochastic fluctuations between external stimuli and epigenetic modifications are at the origin of this expression control.

Moreover, temperature is a key factor in the growth and development of plants. The reaction of a plant at room temperature depends on a variety of factors, including the stage of development. Indeed, an in vitro study revealed that temperature had a significant impact on the biosynthesis of PA in Cabernet Sauvignon grape berries during development by increasing the expression of key candidate genes [87]. Indeed, it is known that temperature affects metabolism; mainly by affecting the action of enzymes responsible for synthesis. Temperature can act negatively (denaturation) or positively (catalysis). However, temperature has also been shown to play an important role in regulating gene expression. Indeed, temperature can predispose a plant to the expression or repression of particular genes by affecting the epigenetic mechanisms that control gene expression. This suggests that depending on the temperature, there will be a synthesis of different metabolites.

Based on all of the aforementioned studies, it can be confirmed that the synthesis of polyphenols by berries is only a plant’s response to various environmental challenges, involving hormonal and genetic status, through modification of polyphenol biosynthetic gene expressions, which depend on cultivar, tissue type, and developmental stage (Table 1).

## 3. Pharmacokinetic Aspects of Berry-Derived Polyphenols

### 3.1. Bioavailability and Metabolism of Berry-Derived Polyphenols

In the last few decades, a particular focus has been put on the investigation of the bioavailability and metabolism of polyphenols as an important step toward a better understanding of the molecular mechanisms underlying their biological activities. In fact, a rising trend in the use of berry-derived polyphenols as functional foods has been reported in several dietary human studies [117,118]. However, these phenolic compounds appear to be poorly absorbed in the gastrointestinal tract (GIT). This low bioavailability has been shown to be due to many factors. Generally, the chemical structure of polyphenols (e.g., glycosylation, esterification, polymerization, and molecular weight) greatly influences their bioavailability, absorption, and metabolism in the GIT [119,120]. 

Polyphenols such as phenolic acids and flavonoid aglycones are readily absorbed in the small intestine. However, phenolic compounds are mainly present in berry fruits in ester, glycoside, or polymer forms (e.g., anthocyanins, ellagitannins, and flavan-3-ols), thus requiring hydrolysis by intestinal enzymes or by the gut microbiota before they can be absorbed.

The metabolic process occurring in the small intestine mainly involves lactase phlorizin hydrolase (LPH), a β-galactosidase enzyme present in the brush border membrane of enterocytes [121]. This enzyme hydrolyses phenolic glycosides to aglycones, which may then enter the intestinal cells by passive diffusion due to their increased lipophilicity [8,122,123]. However, other investigations demonstrated that glycosides could be alternatively transported by the sodium-dependent glucose transporter (SGLT1) into enterocytes, where they undergo hydrolysis by a cytosolic glucosidase [124,125]. It has been reported that both enzymes may be involved, but their relative mechanisms on various phenolic glycosides need further clarification [120].

In the enterocytes, the resulting aglycones are subjected to phase I metabolism (oxidation, reduction, or hydrolysis), and thereby passively diffuse into the liver [1]. The polyphenols and their unabsorbed fractions in the small intestine then reach the colon, where they are transformed by the gut microbiota into several smaller metabolites through the opening of the heterocycle at specific points according to their molecular structure [120,126,127] (Appendix A).

The composition of the gut microbiota has been shown to substantially influence the bioavailability of polyphenols and, consequently, may affect their resulting metabolites [19,21].

The absorbed aglycones reach the liver and once in the hepatocytes, these metabolites undergo phase II metabolism, in particular conjugation, forming sulfate, glucuronide, or methylated derivatives, through the respective action of sulfotransferases, uridine 5′-diphospho-glucuronosyltransferases, and catechol-*O*-methyltransferase [120,127,128]. The candidate metabolites are then released into the systemic circulation for distribution to other organs. There is also an efflux of some of the metabolized polyphenols into the intestinal lumen via ATP-binding cassette (ABC) transporters such as multidrug-resistance proteins (MRPs) and P-glycoprotein (P-gp) [129]. The elimination of polyphenol metabolites occurs via either the biliary or the urinary route [128,130,131].

### 3.2. Pharmacokinetic of Phenolic Acids and Subgroups

Phenolic acids are considered to be one of the main phenolic compounds found in berry fruits, especially chokeberry (96 mg/100 g), blueberry (85 mg/100 g), and sweet rowanberry (75 mg/100 g) [132]. Although dietary phenolic acids have been shown to be responsible for broad health effects in the human body [133,134,135], their bioavailability has not gained huge attention compared to that of flavonoids such as anthocyanins and flavonols. Therefore, further studies on the bioavailability and metabolism of phenolic acids are required to improve in depth our knowledge of their bioactivity.

Generally, phenolic acids have been shown to be readily absorbed in the GIT compared to other phenolic groups such as flavonoids due to their small molecular weights.

Acids such as *p*-coumaric acid, *trans*-cinnamic acid, and ellagic acid extracted from strawberry were found to be greatly absorbed in the small intestine [136]. This could indicate that phenolic acids, particularly ellagic acid, may be included in most of the health benefits of strawberries. Furthermore, using a simulated stomach and duodenum digestion, Burgos-Edwards et al. [137] showed that chlorogenic acid, neochlorogenic acid (3-caffeoylquinic acid), and caffeic derivatives of chokeberry were not altered by stomach digestion, whereas pancreatic digestion affected the level of soluble recovery of these phenolic acids. Indeed, neochlorogenic acid was decreased by approximately 28%, while chlorogenic acid was increased by 24%. This fact could be attributed on one hand to the low chemical stability of phenolic acids under the higher pH (6.5–7.5) of the intestinal compartment, while on the other hand to the presence of a pancreatic and biliary enzymatic mixture. In contrast, another study reported the low stability of phenolic acids present in the genus Ribes after gastric digestion [137]. In addition, an in vitro combination of luminal digestion and intestinal human colon carcinoma cell line (Caco-2) models was used to clarify the mechanism of absorption and digestion of phenolic compounds from chokeberries (*Aronia melanocarpa*). This study showed that chokeberry phenolic acids appeared to be relatively stable in the small intestine, while they were extensively metabolized in the colon, especially in the ascending part. Generally, the basal recovery of caffeic acid (21%) was higher than that of chlorogenic acids (13%). Chlorogenic acid was broken down throughout the various phases of digestion, while quinic acid was significantly increased, suggesting that quinic acid may be produced from chlorogenic acid metabolism [138].

Moreover, the bioavailability of berry phenolic acids was also investigated in vivo using several animal models in order to understand and clarify their absorption and metabolism, and therefore their effects on health. However, these studies did not report data on the bioavailability of phenolic acids in great detail. Wu et al. [139] reported that after consumption of black raspberry (BRB) powder by pigs, the five phenolic acids (protocatechuic, *p*-coumaric, caffeic, ferulic, and 3-hydroxybenzoic acids) originally present in the BRB were all recovered in the GIT at different concentrations. Protocatechuic acid and 3-hydroxybenzoic acid were recovered at higher proportions than their initial presence in the BRB, indicating the production of these compounds throughout the GIT. Additionally, phenolic acids such as homovanillic acid and homoprotocatechuic acid, which were not present in the BRB, were detected in the cecum and/or colon [139]. These compounds were identified as colonic metabolites of the gut microflora. Some claimed that phenolic acids could be produced from the microbial metabolism of other phenolic compounds, mainly anthocyanins. However, in this study, the authors showed that phenolic acids accounted for only 6.31% of anthocyanin degradation, indicating generation of phenolic compounds from miscellaneous sources in the GIT [139]. These findings did not agree with claims that phenolic acids are the major catabolites of anthocyanin breakdown. In a study using rat models, urinary excretion of blueberry, cranberry, or black raspberry polyphenols showed that phenolic acids were excreted in the urine in both conjugated and free forms [140]. The anthocyanin-rich BRB led primarily to the production of 3-hydroxybenzoic, 3-hydroxycinnamic, and 3-hydroxyphenylpropionic acids, while 3,4-dihydroxycinnamic, chlorogenic, and ferulic acids were found to be the blueberry metabolites initially rich in proanthocyanidins [140].

### 3.3. Pharmacokinetic of Flavonoids and Their Subgroups 

As mentioned above, anthocyanins are the most abundant flavonoids in berries. Indeed, most studies on the bioavailability of berry-derived polyphenols focus on these compounds. These include studies using feeding animal models, in vitro gastrointestinal digestion, and volunteer studies. The large heterogeneity in the chemical structures of anthocyanins may influence their bioavailability, absorption, and metabolism, and thereby their potential health effects [18,19]. Anthocyanin aglycones can be absorbed in the stomach and rapidly appear in the bloodstream in their native form [1,8]. These molecules are characterized by short elimination half-lives (less than 2 h) and quick urinary clearance [141]. It has been shown that some of these anthocyanin aglycones can reach the colon, and then appear in the serum as cleaved metabolites of microbial metabolism, mainly phenolic acids [142]. The presence of sugars and acetyl groups affects the stability of anthocyanins with respect to pH, and therefore their absorption in GIT. In general, berry anthocyanins have been shown to be stable and resistant to the acidic conditions of gastric digestion (pH = 2) [1,120,143]. Indeed, experimental in vitro studies have shown that the total anthocyanin content of raspberry, chokeberry, mulberry, and blueberry remained relatively stable without major modifications in terms of composition in gastric digestion conditions (pH = 2 and the presence of pepsin for 2 h) [144,145]. However, it has been reported that under a gradual increase in pH in the stomach compartment, the contents of anthocyanin cyanidin-3-glucoside and cyanidin-3-rutinoside of Chilean currants (*Ribes* genus) was drastically decreased (about 80%) [137]. Unlike the stomach, the intestinal environment has a neutral pH (6.5–7.5), and anthocyanins are considerably less stable there and can be degraded naturally to chalcones [137,143]. Under in vitro gastrointestinal conditions, Bermúdez-Soto et al. [146] reported that the half of chokeberry anthocyanins were recovered in the duodenum, suggesting the cleavage of these compounds to chalcone forms. However, a higher stability has been noticed for blueberry acylated anthocyanins and delphinidin- and malvidin-acetoyl-glucosides [144]. Anthocyanins have been shown to exist as a mixture of four equilibrium molecular species, including the basic flavylium cation (AH+), quinoidal bases, carbinol pseudobase, and chalcone pseudobase forms [147]. The low recovery of anthocyanins might be related to the conversion of the flavylium cation to chalcone pseudobase forms under intestinal pH conditions (Appendix A).

Liang et al. [48] reported that the bioaccessibility of mulberry anthocyanins was highly decreased after intestinal digestion. Only 0.5% of anthocyanins were recovered in the absorbed material, while 4.58% remained in the small intestine and was then released into the colon. Similarly, using a dialysis membrane, the bioaccessibility of the principal blackberry anthocyanin, cyaniding-3-*O*-glucoside, was about 1.8% [148]. This low bioaccessibility could be attributed to the low stability of the compounds under intestinal conditions. Colonic fermentation of anthocyanins leads to the formation of various metabolites, among which phenolic acids predominate. Anthocyanin metabolism by colonic microbiota has been reported to produce gallic, syringic, and *p*-coumaric acids, while syringic acids appear as a major metabolite of malvidin-3-glucoside metabolism [149]. In a recent study, Kim et al. [150] showed that after in vitro gastrointestinal digestion of chokeberry and mulberry anthocyanins, the phenolic acids, including *p*-coumaric acid, protocatechuic acid hexoside, caffeoylquinic acid, coumaroylquinic acid, feruloylquinic acid, and numerous cyanidin conjugates, were newly formed. These candidate metabolites are mainly produced from the decomposition of cyanidin-glycosides [150]. In humans, protocatechuic acid has been reported as the main metabolite of cyanidin-3-*O*-glucoside [151]. The composition of the gut microbiota significantly affects the metabolization of anthocyanins, and consequently their produced metabolites. Mulberry anthocyanis was found to be highly converted by probiotic strains such as *Lactobacillus plantarum* and *Streptococcus thermophilus*, which induced hydrolyzation of cyanidin-3-glucoside and cyanidin-3-rutinosid. Chlorogenic acid, cryptochlorogenic acid, and caffeic acid were the main produced metabolites [152]. 

In animal studies, berry anthocyanins have been shown to be absorbed mainly in their native glycosidic form, and then rapidly reach the systemic circulation (0.25–2 h). A previous study reported that after a single oral administration of blueberry anthocyanins in rat models, the plasma concentrations reached the peak level (2–3 µg/mL) in just 15 min, and then quickly declined after around 2 h. Moreover, other studies have investigated the fate of black raspberry anthocyanins in three different segments of the GIT (small intestine, cecum, and colon) of pigs. The authors showed that the molecular structure of anthocyanins greatly influenced their GI recovery. Approximately 98% of the cyanidin-3-glucoside disappeared from the GIT within 4 h of administering the anthocyanins, indicating their direct absorption. However, the cyanindin derivatives with more complex glucosides such as sambubiose or rutinose showed a higher recovery in the colon, and were responsible for most of the identified metabolites in the urine [139].

Generally, the bioavailability of berry anthocyanins was reported to be low in the human body, and cyanidin-3-glucoside was the main anthocyanin detected in plasma and urine. Mueller et al. [118] showed that after oral intake of blueberries by healthy volunteers, anthocyanins were rapidly absorbed, and their relative bioavailability was very low in plasma (0.02%) and urine (0.03%). The authors also reported that anthocyanins were mainly absorbed in the form of glucuronides, with a predominance of the glucurone forms of malvidin (28%) and peonidin (46%) [118]. Furthermore, after consumption of a single dose of black currant concentrate (3.57 mg cyanidin-3-glucoside/kg BW), black currant anthocyanins were absorbed directly, distributed in the circulatory system, and then excreted in urine in glycosylated forms (intact forms) [153]. The low bioavailability of berry anthocyanins was also reported in a human study conducted by Murkovic et al. [154]. After taking a 180 mg dose of elderberry anthocyanins, the maximum plasma concentration was very low (35 ng/mL). Additionally, Milbury et al. [155] showed that cranberry anthocyanins were poorly absorbed and rapidly cleared from plasma. The plasma concentrations of the anthocyanins varied between 0.56 and 4.64 nmol/L. The urinary recovery ranged from 0.078% to 3.2%, with notable interindividual variations. Overall, several berry-feeding studies reported that most anthocyanins were excreted within the first 4 h (29), and total recovery from black currants, elderberries, boysenberries, and blueberries varied between 0.02 and 0.37% [155,156,157,158].

On the other hand, flavonol quercetin from berries has been reported to be readily bioavailable. Indeed, after ingesting black currant juice, the concentration of quercetin in the plasma was higher compared to that of the control group (baseline). In volunteer subjects that consumed 100 g per day of a berry mixture containing black currants, bilberries, and lingonberries, the plasma quercetin levels were significantly increased (from 30% to 50%) [159].

### 3.4. Pharmacokinetics of Other Polyphenolic Compounds 

Ellagitannins (ETs), a class of hydrolyzable tannins, are bioactive phenolic compounds that are mainly found in red fruits, including berries such as raspberries, strawberries, and blackberries [40,160]. These molecules are degraded spontaneously into ellagic acid during gastrointestinal digestion, which in turn is metabolized into various urolithins by a specific gut microbiota (Figure 3). These urolithins can further undergo phase II metabolism in the small intestine and liver before entering the circulatory system, producing derivatives of glucuronides and sulfates [161,162]. 

Van de Velde et al. [148] studied the in vitro bioaccessibility of blackberry ellagitannins at simulated gastrointestinal and colonic levels. The intestinal bioaccessibility was about 1% of the main ETs lambertianin A and C, while this value was higher than 14.9% for ellagic acid, due to its release from ET breakdown. Lambertianin A and C were not detected after colonic fermentation, suggesting hydrolysis of these compounds to ellagic acid in colonic digestions. However, the colonic bioaccessibility of ellagic acid was less than 10%, indicating further conversion of this compound into small molecules [148]. Indeed, it has been shown that ellagic acid was rapidly metabolized into urolithin B and urolithin C by colonic microbiota after in vitro colonic fermentation [145]. In addition, in vitro anaerobic incubation of ETs with fecal suspensions showed metabolization to ellagic acid and various urolithins. Thus, 80% of the added ellagic acid were metabolized into urolithins in the fecal suspensions [163]. Furthermore, in a feeding study with rat models, ETs were found to be rapidly cleared in the stomach and undetectable in the plasma, urine, feces, duodenum, jejunum, and other parts of the GIT within one hour of raspberry ingestion [164]. This event may be related to the fact that the acidic conditions of the stomach compartments led to the breakdown of sanguiin H-6 and lambertianin C initially present in raspberry juice. The recovery of ellagic acid in the stomach after 1 h was around 9.6% of the amount present in the raspberry juice, while only traces of ellagic acid were detected after 2 h. However, no ellagic acid was detected in rat organs/tissues and fluids (plasma, urine) collected over a 24 h period after raspberry ingestion [164]. 

Generally, ETs are highly metabolized by the human gut microbiota, producing dibenzopyran-6-one metabolites, mainly urolithin A and its monohydroxy analogue, urolithin B. In a human feeding study conducted by González-Barrio et al. [163], it was shown that urolithin was excreted in the urine mainly as O-glucuronides, indicating the conversion of ETs into urolithins by the colonic microbiota, while glucorinidation occurred in the colon and/or after absorption in the liver.

González-Barrio et al. [165] reported the metabolic fate of ETs and ellagic acid after consumption of 300 g of raspberries by human volunteers. This study showed that the recovery of ellagic acid in ileal fluid was approximately 241%, with hydrolysis of ellagitannins in the gastrointestinal tract. Urinary excretion of ellagic acid and an ellagic acid-*O*-glucuronide was lower at 1% of intake, whereas no intact or conjugated forms of ETs were detected in urine. Qualitative and quantitative variations have been noted in the urolithin metabolic phenotypes (metabotypes) of individual volunteers [165]. In the same context, another human investigation that evaluated the fate of ETs from strawberry reported large interindividual variabilities of urolithin metabolism. Urolithin was found only in 68% of cases, with significant interindividual variability. Urolithin A and iso-urolithin A glucuronide were produced by 13 subjects, while urolithin B glucuronide was found in the plasma of 6 subjects, 1 of whom showed only urolithin B [162]. This event may have been related to each individual’s ability to produce specific metabolites and to the genetic polymorphism of metabolizing enzymes on the one hand, and on the other hand to the gut microbiota composition [1]. However, studies that evaluated the action of specific microorganisms on the metabolism of berry-derived polyphenols are still limited due to the high complexity of berry matrices, as well as the composition of gut microflora. 

## 4. Effects of Berry-Derived Polyphenols on Gut Microbiota and Related Pathogenic Bacteria

The gut microbiota is a complex and diverse microbial community composed of a broad spectrum of species. The gut microbiota includes bacteria, viruses, fungi, archaea, and protozoans. Approximately ~10^14^ bacterial cells, which is 10 times more than the total number of somatic and germ cells in the human body, are present in the gastrointestinal tract, but they are mostly located in the large gut [166,167]. *Bifidobacterium*, *Clostridium*, *Eubacterium*, *Peptococcus*, *Ruminococcus*, and *Peptostreptococcus* are the predominant bacteria of colonic microbiota, while the subdominant species comprise *Escherichia*, *Klebsiella*, *Enterococcus*, *Enterobacter*, *Proteus*, and *Lactobacillus* [168]. They support a wide range of physiological functions through various enzymatic and metabolic processes that impact host nutrition and health [168]. Gut microbiota can hydrolyze glycosides, sulfates, glucuronides, amides, lactones, and esters through the action of enzymes such as β-glucosidase, sulfatase, β-glucuronidase, α-rhamnosidase, and esterases. Other reactions catalyzed by intestinal enzymes are decarboxylation (descarboxylase), isomerization (isomerase), reductions (reductases, hydrogenases), dehydroxylation (dehydroxylase), demethylation (demethylase), and aromatic ring-cleavage [169,170]. Intestinal bacteria constitute a mechanical and immunological barrier between a host and the environment [171]. The main important functions of the gut microbiota for human hosts are in nutrient processing; modulation of the intestinal immune response [172]; protection against host radiosensitivity [173,174]; and cardiovascular, gastrointestinal, and other various disorders [175,176]. Microbial colonization of human gastrointestinal tract by environmental bacteria begins immediately after birth. The composition of gut microbiota is relatively simple in early infancy; however, it becomes more complex with age, with a considerable degree of variability between different individuals [177]. Although more than 1000 types of bacteria are present in the human intestine, only 150 to 170 species are common to individuals [178]. Bacteria present in the intestinal lumen can be classified according to their degree of pathogenicity [179]. Three classes of bacteria can be defined: beneficial, potentially pathogenic, and pathogenic species. Beneficial effects of bacteria include inhibiting the growth of pathogenic bacteria, the breakdown and fermentation of food ingredients, production of vitamins, stimulation of immune functions, and stimulation of food tolerance. The pathogenic effects are food intolerance, infections, and inflammation [179]. Potentially pathogenic species belong to the normal microflora of the gut, but may become pathogenic when present in large numbers; these include *Escherichia coli*, *Enterobacteria*, *Streptococci*, *Enterococci*, and *Bacteroides*.

A gut microbial profile in which beneficial bacteria predominate over pathogenic bacteria is desirable, as it provides other health benefits to the human host and promotes immune functions [171]. However, various endogenous and exogenous factors, as well as diet, age, genetics, immunological status, and antibiotic therapy, can disrupt the intestinal microbiota balance, and consequently, the appearance of gastrointestinal disorders such as inflammatory bowel disease, irritable bowel syndrome, and antibiotic-associated diarrhea [180,181]. Polyphenol components found as secondary metabolites in vegetables and fruits have considerable beneficial health properties, including the reduced risk of various chronic illnesses such as type 2 diabetes, coronary heart disease, neurodegenerative diseases, and certain cancers [182,183]. Moreover, there is growing evidence that modest, long-term intakes of polyphenols can modulate the gut microbiota and contribute to a beneficial microbial environment that can remarkably improve human health. It is now well established that polyphenols modify many aspects of gut composition, such as the abundance of *Bifidobacterium*, *Proteobacteria*, *Actinobacteria*, *Deferribacteres*, *Lactobacillus*, *Bifidobacterium*, *Helicobacter*, *Desulfovibrio*, *Adlercreutzia*, *Prevotella*, and *Flexispira* [184,185]. Preclinical and clinical experimental evidence suggests that polyphenols play an important role in increasing the beneficial gut bacteria, such as Akkermansia muciniphila, and decreasing the pathogenic gut microbiota [186].

Modulation of the gut microbiota by berry polyphenols is shown in Table 2. The influence of polyphenols on intestinal microbiota growth and metabolism depends specifically on the polyphenol structure, their metabolism in the gut, and the microorganism strain [183,187,188]. For instance, Gram-negative bacteria are less resistant to polyphenols than Gram-positive bacteria, possibly due to observed differences in their wall compositions [188]. *Bacteroidetes* and *Firmicutes* are the two most abundant bacteria in human and mice intestinal microbiota. High-fat diets are known to promote the growth of *Firmicutes* and reduce the abundance of *Bacteroidetes*. A high *Firmicutes*/*Bacteroidetes* ratio is associated with several intestinal disorders, including obesity [189]. In mice given a high-fat diet (without berry extracts), this change in the *Firmicutes*/*Bacteroidetes* ratio was observed [190,191]. However, a *Lonicera caerulea* berry supplement reversed this trend, and the ratio decreased [191]. In mice fed a high-fat diet, grape polyphenols led to promotion of the growth of *A. muciniphila* and a reduction in the *Firmicutes*/*Bacteroidetes* ratio [192]. This study also showed that consumption of grape polyphenols attenuated several effects of a high-fat diet, including weight gain, glucose intolerance, adiposity, and serum inflammatory markers (interleukin (IL)-6, tumor necrosis factor (TNF)-α, and lipopolysaccharide) [192]. The consumption of black currant and blueberry significantly increased *Actinobacteria* and *Bacteroidetes* populations, two obligate anaerobes. Furthermore, a potential decrease in intestinal oxygen tension caused by the addition of berry-derived anthocyanins could stimulate the growth of many oxygen-sensitive species [187]. Similar results were seen in obese human volunteers. The consumption of an anthocyanin mixture derived from blueberry, black currant, and black rice associated with a prebiotic mixture (inulin) by obese subjects increased *Bacteroidetes* and decreased *Firmicutes* and *Actinobacteria* populations, thus reducing the *Firmicutes*/*Bacteroidetes* ratio [193].

Several studies have shown that the administration of black currant extracts in animals reduces the *Bacteroides* population. A diet containing raspberry considerably reduced *Clostridium* spp., in mice [194]. Consumption of lingonberries by atherosclerosis-prone mice decreased *Oscillospira* and *Musispirillum*, while the abundance of *Bacteroides*, *Parabacteroides*, and Clostridium increased [195]. The same researchers also demonstrated a positive correlation between the *Musispirillum* populations and the number of plaques. Therefore, lingonberry can be beneficial in vascular disorders [195]. In rats force-fed with black currant extracts, a significant increase in the bifidobacteria and lactobacilli populations and a remarkable decrease in the number of bacteroides and clostridia were recorded [196].

In clinical studies, consumption of blueberry by healthy volunteers resulted in an increase in *Bifidobacterium* spp. and *Lactobacillus acidophilus* [197]. The analysis of the gut microbiota of healthy volunteers consuming black-currant-based products showed a decrease in *Bacteroides* and *Clostridium* spp. [198]. The consumption of *Schisandra chinensis* fruit by obese volunteers resulted in increased *Bacteroides*, *Bifidobacterium*, *Roseburia*, and *Prevotella* populations [199].

The modulation of the intestinal microbiota by in vitro assays was also assessed. An evaluation of the antimicrobial ability of cranberry extracts showed significant antimicrobial effects on various human colonic bacteria, such as *Bacteroides* and *Prevotella* populations [200]. Similar effects against *Bacteroides* and *Prevotella* were observed when crop populations were treated with anthocyanin extracts of *Lycium ruthenicum* Murray and açai pulp [201,202].

Recent discoveries suggested a variety of potential mechanisms for modulating polyphenols in gut bacteria. For instance, various polyphenolic compounds can bind to microbiota cell membranes, thus disturbing the membrane function, and therefore inhibiting cell growth [203]. Catechin, a polyphenol widely present in many berries, acts on different bacterial strains (*Escherichia coli*, *Salmonella choleraesis*, *Staphilococcus aureus*, *Bacillus subtilis*, *Klebsiella pneumonie*, *Bordetella bronchiseptica*, *Pseudomonas aeruginosa*, and *Serratia marcescens*) by generating hydrogen peroxide and by disrupting the permeability of the bacterial membrane [204]. It may also be related to the ability of these compounds to bind to bacterial adhesions, and therefore disrupt the availability of cell surface receptors [205]. Some research has indicated that polyphenol compounds can attack the cell wall and cell membrane, thus destroying their permeability barrier and engendering the release of intracellular constituents such as ribose and sodium glutamate. Furthermore, the polyphenols might interfere with nutrient uptake, nucleic acid and protein synthesis, electron transport, and enzyme activity, leading to bacterial growth inhibition [206,207].

There is substantial evidence that consumption of berry polyphenols modulates the colonic microbiota and protects human health [1,197,222]. A significant modulation of the gut microbiota can be noted at the level of the two most abundant phyla in the human and murine intestinal microbiota—*Firmicutes* and *Bacteroidetes*. High-fat diets have been found to reduce the growth of *Bacteroidetes* and promote the abundance of *Firmicutes*. A ratio in favor of *Firmicutes* has been linked to various conditions, such as obesity [189]. In mice consuming high-fat diets (without berry polyphenols), this change in the Firmicutes/Bacteroidetes ratio was detected [190,191]. In obese human subjects, consumption of a mixture of anthocyanins from blueberry, black currant, and black rice remarkably increased the *Bacteroidetes* population and significantly reduced the *Actinobacteria* and *Firmicutes* populations, therefore decreasing the *Firmicutes*/*Bacteroidetes* ratio [193]. In vitro investigations revealed the antimicrobial effects of cranberry and grape seed polyphenols against *Bacteroides*, *Prevotella*, and *Blautiacoccoides*–*Eubacterium rectal* [200]. Other in vitro studies [201] and [202] reported a significant antimicrobial effect of the anthocyanin extracts from açai pulp and *Lycium ruthenicum,* also called Russian box thorn, against *Bacteroides* and *Prevotella* populations. The consumption of a blackberry anthocyanin-rich extract by mice fed a high-fat diet may have modulated the gut microbiota composition and counteracted some of the features of dysbiosis induced by the high-fat diet by decreasing *Rumminococcus* and increasing *Oscillobacter* and *Sporobacter* [208]. The administration of lingonberries to atherosclerosis-prone mice remarkably decreased *Musispirillum* and *Oscillospira*, while *Parabacteroides*, *Bacteroides*, and *Clostridium* were increased. This investigation also demonstrated a positive association between the relative abundance of *Musispirillum* and the number of plaques. Therefore, lingonberry can also reduce vascular disease [195].

In human studies, the consumption of *Schisandra chinensis* fruits (called magnolia berries) by obese volunteers showed an increase in *Bacteroides*, *Bifidobacterium*, *Prevotella*, and *Roseburia* [199]. An analysis of the gut microbiota of volunteers consuming black-currant-based products showed a decrease in *Bacteroides* and *Clostridium* sp. [198].

The antimicrobial effects of berry polyphenols against pathogenic bacteria have been extensively investigated [10,188]. Using agar diffusion assays on pathogenic bacteria of the human intestine, the berries showed a more significant inhibitory effect on Gram-negative bacteria than on Gram-positive bacteria [188]. Using the same test, Puupponen-Pimiä et al. [10] showed that ellagitannins of cloudberry and raspberry strongly inhibited *Staphylococcus* species, whereas a *Listeria* culture was unaffected by the whole berries and phenolic extracts, except for cranberry [10]. In the same sense, açai berry showed a significant antimicrobial effect against *Clostridium histolyticum*, a pathogen associated with tumor promotion and inflammatory bowel disease [201]. The administration of *Lonicera cerula* fruits (commonly known as magnolia berry) to mice fed a high-fat diet markedly decreased *Staphylococcus* species [191]. In another in vivo model, Paturi et al. [210] demonstrated the ability of blueberries to positively modify the composition of the gut microbiota by decreasing the *Enterococcus* spp., *Clostridium perfringens,* and *E. coli* populations. 

In addition to berry-derived polyphenols’ effects against pathogenic gut microbiota, they can also affect the composition of nonpathogenic gut bacterial species. Thus, it has been reported that certain berry polyphenols can promote the growth of beneficial bacteria, and at the same time suppress that of pathogenic species [168]. This mostly refers to blueberry polyphenols, but other berry bioactive compounds (buckthorn berries, sea buckthorn, and juçara pulp) can act similarly [202,215,219,220]. It has been observed that the daily consumption of blueberry can modulate intestinal bacteria and increase the number of other beneficial species [197]. Evaluation of the effects of berry polyphenols on beneficial gut bacteria has been widely performed in vivo [149,202,215,219,220]. The growth of bacteria genera can be positively influenced by the phenolic compounds extracted from berries, and can provide some health benefits to the host through a prebiotic-like effect. The growth of *Bifidobacterium* and *Lactobacillus* spp. bacteria was enhanced in batch cultures due to the properties provided by the malvidin-3-glucoside extract [149]. Berry extracts and juçara pulp have different effects on bacteria genera growth. Extracts of sea buckthorn berry, blueberry, and *Lycium ruthenicum* berry have shown increased *Lactobacillus* spp. growth, as well as bifidogenic effects. On the other hand, juçara pulp showed an increased growth of *Bifidobacterium* and no growth difference for Lactobacillus [202,215,219]. The mechanisms of the in vitro modulation remain unclear, and further studies must be carried out. Contrary to the increased production of short-chain fatty acids (SFCAs), multiple studies reported the effects of berry extracts and supported the prebiotic-like properties of the extract [202,215]. Another property of sea buckthorn juice was reported by Attri and Goel [220]. The juice had an inhibitory effect for 48 h on beneficial bacteria populations. This short-term inhibition stopped the bacteria populations’ growth before they began to increase immediately or stabilized. This stabilization might have been an adaptation phase before proliferation.

In vivo studies on the effects of berry-derived polyphenols in the gut have been widely performed and several investigations have demonstrated effects of berries in restoring a balanced gut microbiota [191,210]. The administration of honey berry (*Lonicera cerula* L.) to mice fed a high-fat diet remarkably decreased the *Lactobacillus* population [191]. Blueberry consumption by the inflammatory bowel disease (IBD) mouse model also decreased Lactobacillus spp., but no effect was reported for *Bifidobacterium* [210]. In rats fed a high-fat diet, the addition of *juçara* pulp contributed to the remodeling of their gut microbiota by an accumulation of *Bifidobacterium* spp. [216]. The supplementation of juçara pulp in the diet of mother mice, in which the population of Bifidobacterium in the colon of their offspring was reduced, restored the population of *Bifidobacterium* in the offspring, suggesting the protective effects of this fruit in maternal nutrition [223]. Strawberry supplementation in a diabetic mouse model contributed to an increase in the *Bifidobacterium* population, the level of which was higher in healthy mice than in diabetic animals, suggesting protective effects of berry-derived polyphenols in the gut [221].

In clinical trials, consumption of a blueberry drink for 6 weeks significantly increased the *Bifidobacterium* population [213]. In healthy volunteers who consumed black currant products, an increase in both the *Bifidobacterium* and *Lactobacillus* populations was observed [198]. *Magnolia* berry (*Schisandra chinensis*) consumption by obese volunteers led to an increase in the *Akkermansia* population [199].

The modulatory effects of berries on the gut microbiota are likely due to the berry constituents that can reach the intestine and modulate the microbiota by increasing the growth of some species and decreasing others [199].

## 5. Therapeutic Applications of Berry-Derived Polyphenols

Berries have widely been studied for their uses in various therapeutic applications, as well as their roles in promoting human health and preventing disease [224], which are mainly attributed to their content of phytochemicals and nutrients, including polyphenols, in emphasizing anthocyanins, antioxidants, vitamins, minerals, and fiber. Due to the high levels of polyphenols in berries [225], in vitro and in vivo studies have shown that berries provide an important range of biological activities [224], which may be associated with a reduced incidence of disorders induced by reactive oxygen species (ROS) [226], such as beneficial effects on inflammatory-related pathologies, various types of cancers and metabolic disorders, as well as antioxidant properties [6,227].

### 5.1. Effects on Gut Inflammation 

The gut microbiota plays a crucial role in immunological, developmental, and nutritional host functions, and therefore has a profound impact on human health [228]. Recent investigations suggest that an individual’s diet can strongly influence changes in the microbiota that, in turn, affect intestinal permeability and lead to a chronic/low-grade inflammatory state and endotoxemia [229]. It is well known that dietary factors such as fat, fructose, and alcohol can alter gut flora and gut permeability, allowing gut-derived toxins to cross the intestinal barrier and activate hepatic cells, which are responsible for overproducing inflammatory cytokines, with the main mechanism responsible being an increase in NF-κB (nuclear factor kappa light-chain enhancer of activated B cells), subsequent gut/hepatic inflammation, and ultimately, systemic inflammation and organic damage [229,230].

Crohn’s disease and ulcerative colitis are two chronic IBD disorders that affect the intestine [1]. Results from in vitro, in vivo, and human studies suggested that polyphenols, especially berry anthocyanins, due to their antioxidant and anti-inflammatory effects, may modulate the colonic microbiota and contribute to the alleviation of symptoms of intestinal inflammation via the modulation of proinflammatory cytokines [228].

#### 5.1.1. In Vitro Studies

The mechanisms underlying the potential therapeutic action of blueberry anthocyanins have been evaluated in vitro. As identified by Triebel et al. [231], single anthocyanins and extracts of bilberry (*Vaccinium myrtillus* L.) significantly inhibited the expression of proinflammatory mediators, including TNF-α, interferon γ-induced protein 10 (IP-10), interferon-inducible T-cell α chemoattractant (I-TAC), soluble intercellular adhesion molecule-1 (sICAM-1), and chemokine growth-related oncogene-alpha (GRO-α). These results suggested that the investigated anthocyanins could be useful in the treatment of IBD as nutritional supplements. Another study performed in vitro by Esposito et al. [232] determined the capacity of major polyphenols present in wild blueberries to protect murine RAW 264.7 macrophages from lipopolysaccharide (LPS)-induced inflammation (LPS plays a crucial role in the relationship between the gut microbiota changes and inflammation). The results of this study showed that malvidin-3-glucoside had the highest anti-inflammatory activity than epicatechin or chlorogenic acid in reducing proinflammatory gene expression via inhibition of the NF-κB pathway. 

#### 5.1.2. In Vivo Studies

The beneficial effect of blueberry on gut inflammation has been studied in rat and mouse models with induced colitis. Pervin et al. [233] showed that blueberry extract (BE) could attenuate the colitis disease activity index (DAI) in mice supplemented with 50 mg BE/kg for 7 days. BE markedly decreased myeloperoxidase (MPO) activity, as demonstrated in several studies [234,235]. MPO is a marker of the presence of neutrophils in the intestinal mucosa, which promotes the formation of oxidants, leading to the onset of inflammation [1]. In addition to this, levels of malondialdehyde, prostaglandin E_2_ (PGE_2_), interleukin-1β (IL-1β), cyclooxygenase (COX)-2, and NF-κB were reduced following the anti-inflammatory action of BE, the inhibition of NF-κB pathway, and downregulation of proinflammatory cytokines [233]. Research performed by Wu et al. [235] revealed that treatment of experimentally induced colitis in mice with anthocyanin extract of blueberry restored IL-10 excretion and reduced levels of nitric oxide (NO), MPO, interferon (IFN)-γ, IL-12, and TNF-α. Osman et al. [234] showed that blueberry alone and in combination with probiotic strains reduced the severity of inflammation in an induced colitis model, reduced bacterial translocation, and improved the DAI. Furthermore, a study carried out by Shusong Wu et al. [191] found that *Lonicera caerulea* L. berry polyphenols (LCBP) protected mice that were fed a diet of 0.5–1% of LCBP for 45 days against nonalcoholic fatty liver disease (NAFLD) induced by a high-fat diet by inhibiting the production of mouse inflammatory cytokines, including IL-2, monocyte chemotactic protein-1 (MCP-1), and TNF-α. In addition, this study revealed that LCBP supplementation potentially led to attenuate inflammation in NAFLD through modulation of the gut microbiota, especially the *Firmicutes*/*Bacteroidetes* ratio, using high-throughput 16S rRNA gene sequencing. Other studies that were performed showed that cranberry administration also relieved intestinal inflammation and oxidative stress in colitis induced in mice by the authors [11,190]. Anhê et al. [11] noted that cranberry extract (CE) contributed to a decrease in the expression of TNF-α and COX-2, and to the normalization of the NF-κB/IκB ratio. However, Xiao et al. [11] suggested that cranberry products not only had beneficial effects on colonic MPO activity and production of proinflammatory cytokines, but also on the attenuation of colonic shortening. The mechanism suspected by the researchers was a regulatory effect of the fermented products from the gut microbiota to participate in the beneficial effects of cranberries on ulcerative colitis.

Black raspberries (BRBs) also play a preventive role in the pathogenesis of ulcerative colitis due to their potent anti-inflammatory effect, leading to a decrease in proinflammatory cytokines. As demonstrated by Montrose et al. [236], BRB treatment dramatically suppressed the levels of phospho-IκBα (nuclear factor of kappa light polypeptide gene enhancer in B-cells inhibitor, alpha), TNF-α, IL-1β, COX-2, and prostaglandin E_2_ in an experimental study of induced ulcerative colitis in mice fed a diet containing BRB powder (5 or 10%) for 7–14 days. On the other hand, 5% dry BRB powder for 28 days in induced ulcerative colitis in mice was sufficient to reduce inflammation by various mechanisms, such as a reduction in macrophages and neutrophils, regulation of protein expression (DNA (cytosine-5-)-methyltransferase 3 beta (DNMT3B), histone deacetylases 1 and 2 (HDAC1 and HDAC2), and methyl-binding domain 2 (MBD2)), correcting promoter hypermethylation in the Wnt pathway and preventing phosphorylation of Iκβα in the colon, resulting in inhibition of nuclear localization of NF-κB p65 and its target genes involved in inflammation (COX-2, TNF-α, and IL-1β) [237]. Another type of berry that has been shown to be effective against intestinal inflammation is bilberry, anthocyanins from which had an apparent ability to improve acute and chronic symptoms of IDB in experimental colitis in mice via a reduction in IFN-γ, TNF-α, and cytokine secretion, as well as a remarkable decrease in epithelial cell apoptosis [238].

In conclusion, according to several studies, it seems that when berries were ingested using colitis-induced mouse models, symptoms of colitis were improved. The findings suggested a potential anti-inflammatory effect of berry compounds (polyphenols and anthocyanins), which appear to inhibit activation of the signaling pathways mediated by the transcription factor NF-κB and oncogenic mitogen-activated protein kinase (MAPK), and consequently the downregulation of proinflammatory cytokines [224]. Nevertheless, more consistent studies are required regarding the anti-inflammatory potential of the anthocyanins, which may not be a general effect, but could be a result of the interaction of many factors, such as the relationship with diet, eating habits, the food source, and the inflammatory status [239,240]. This finding was supported by a study conducted by Graf et al. [239] in which the consumption of 15 mg/day of anthocyanin during treatment with grape juice and blueberry did not affect gut-associated immunity and inflammatory status in healthy rats. 

#### 5.1.3. Clinical Trials

Biedermann et al. [241] conducted human trials to investigate the clinical efficacy of an oral anthocyanin-rich bilberry preparation in 13 patients with mild to moderate ulcerative colitis. At the end of the treatment period (6 weeks), 63.4% of patients achieved remission, and a response in 90.9% of patients was associated with a decrease in mucosal inflammation, which resulted in a decrease in fecal calprotectin levels. Parallel to this study, this compound was also tested in vitro and in patients with ulcerative colitis. The anthocyanin-rich bilberry treatment showed inhibition of IFN-γ receptor 2 expression in human THP-1 cells, and colon biopsies from UC patients detected reduced amounts of the proinflammatory cytokines IFN-γ and TNF-α in patients in remission [242]. On the other hand, a clinical trial involving 12 healthy volunteers showed that the consumption of anthocyanin-rich Açai juice and pulp (7 mL/kg of b.w.) after an overnight fast increased the serum level of anthocyanins, which reached a peak of 2321 ng/L 2.2 h after consumption in the form of pulp and 1138 ng/L 2.0 h after juice consumption [243]. 

Bilberry anthocyanin was effective in illustrating the therapeutic potential in patients with ulcerative colitis. The anti-inflammatory mechanisms of blueberry compounds primarily involve the modulation of T-cell cytokine signaling [1].

### 5.2. Effects on Different Types of Cancer 

Cancer remains a major public health problem worldwide and the second-leading cause of death [244]. The incidence of many types of cancer continues to rise, despite many successes in screening, prevention, and treatment [245]. Over the past decade or more, experimental and clinical investigations have provided convincing evidence that berry polyphenols should be effective against multiple targets in cancer development and progression [246]. It is well established that increased ROS production has been revealed in various types of cancers and has been shown to have several effects, such as activating protumorigenic signaling, enhancing cell survival and proliferation, and driving DNA damage and initiating oxidative stress [247]. As antioxidants, berry polyphenols have been shown to exhibit many anticarcinogenic properties, including their inhibitory effects on cancer cell proliferation, tumor generation, metastasis, angiogenesis, and inflammation, as well as inducing apoptosis in cancer cells and interfering in tumor progression [246,248,249]. Additionally, polyphenols can modulate the immune system response and protect normal cells against free radical damage [250]. Moreover, in vivo and in vitro studies around the world have shown that a high intake of diet-derived polyphenols/anthocyanins (or diets containing freeze-dried berries) was associated with a lower risk of cancer, and recent findings indicated that they also exhibit cancer-preventive effects in humans [246,249,251,252].

#### 5.2.1. Colorectal Cancer

Colorectal cancer (CRC) is an important public health problem [253] and the most common cancer in the Western world [254]. Its incidence has increased rapidly with age for unclear reasons until now [255]. It is the third most common cancer in men after prostate and lung cancer, and the second most common in women after breast cancer [256]. Dietary, environmental, and genetic factors play central roles in the pathogenesis of colon cancer [254,257]. It has been established that animal consumption of a high-fat diet may be strongly linked to colonic carcinogenesis, with both the type and amount of dietary fat consumed involved. In addition, IBD, which is related to reactive nitrogen intermediates (RNIs) and ROS [258], is a chronic, immune-mediated disease that affects the gut microbiota, and included Crohn’s disease and ulcerative colitis, which can dramatically increase the risk for colonic dysplasia and development of CRC [236,259]. Berry extracts and anthocyanin compounds have been studied both in vivo and in vitro, as well as in clinical trials, to evaluate their chemopreventive effects in CRC.

##### In Vitro Studies

Six commonly consumed berries and their phenolic compounds, such as anthocyanins, flavonols, flavanols, ellagitannins, gallotannins, proanthocyanidins, and phenolic acids, were tested in vitro to evaluate their anticancer effects in HT-29/ HCT116 colon cancer cells. The results showed the capacity of the six berries (blackberry IC_50_ = 64.60, blueberry IC_50_ = 89.96, raspberry IC_50_ = 187.60, black raspberry IC_50_ = 89.11, cranberry IC_50_ = 121.30, and strawberry IC_50_ = 114.20) to stimulate apoptosis of the COX-2 enzyme expressing colon cancer cell line, HT-29, and an increase in the inhibition of cell proliferation (at a concentration of 200 µg/mL). Nevertheless, black raspberry and strawberry extracts showed the most significant proapoptotic effects against human cancer cells and berry phenolics, with anthocyanins being major contributors to the induction of apoptosis [260]. According to another study, strawberries may have beneficial effects against oxidative stress, as they revealed antitumor effects on CaCo-2 CRC cells in vitro [261].

Cyanidin-3-glycoside, which belongs to the class of anthocyanins found in black raspberry and is chemopreventive against oxidative DNA damage, led to reduced DNA strand breakage in human colon epithelial cells (HCECs), and was effective in inhibiting peroxyl radical-induced cytotoxicity via inhibition of apoptosis and decreasing the G_1_ phase of the cell population in cultured Caco-2 [262,263].

The effective anticancer agents of cranberries and the nature of the chemopreventive compounds in the fruit were the subject of a recent investigation by Neto et al. [264], who tested proanthocyanidin extracts from cranberry fruit (*Vaccinium macrocarpon*) against eight tumor cell lines, including HT-29 colon cells. The results showed inhibition of tumor cell growth in vitro at GI_50_ values ranging from 20 to 80 µg/mL. Ferguson et al. [265] noted that an acidified methanol eluate (Fr6) from cranberry containing phenolics and flavonoids inhibited proliferation with only an intermediate effect on the HT-29 line colon cells (IC_50_ = 168 ± 69 mg/L of Fr6) via a progressive cell-cycle arrest leading to apoptosis. 

All extracts of chokeberry, bilberry, and grape were significantly effective in inhibiting the growth of HT-29 colon cancer cells, with chokeberry having the most potent effect (IC_50_ = 25 μg/mL of chokeberry anthocyanin-rich extracts after 48 h of exposure). In addition, the berry anthocyanin extracts had no adverse effects on nontumorigenic colon NCM460 cells when used at a lower concentration, showing little or no toxicity on normal cells [266].

##### In Vivo Studies 

Shi et al. [267] demonstrated that 2.5%, 5%, or 10% dietary lyophilized strawberries in the AOM/DSS mouse model significantly reduced tumor incidence and expression of proinflammatory mediators (TNF-α, IL-1β, IL-6, COX-2, and iNOS); suppressed nitrosative stress; and decreased the phosphorylation of phosphatidylinositol 3-kinase (PI3-K), Akt (protein kinase B), extracellular signal-regulated kinase (ERK), and NF-κB (Figure 4).

Other works reported that strawberry extracts were shown to inhibit the proliferation of human colon cancer HT-29 cells, primarily by inducing cell apoptosis and the p21^WAF1^ pathway [268]. Strawberries also had beneficial effects against oxidative stress, and revealed potential antitumor effects on CRC HCT-116 cells [269].

Ellagitannins from black raspberry and their derivatives (ellagic acid, urolithin A, and urolithin B) have shown potential anticancer activity in HT-29 colon cancer cells by regulating the cell cycle (upregulation of p21) and apoptosis signaling pathways, as confirmed by activation of caspase 3 and cleavage of poly (ADP-ribose) polymerase (PARP) [12].

A study by Renis et al. [270] assessed the molecular mechanisms responsible for the anticancerogenic potency of both cyanidin chloride and cyanidin-3-*O*-β glucopyranoside derived from anthocyanin on CaCo2 cells. The results showed inhibition of cell growth/proliferation and a decrease in the ROS level by increasing ataxia telangiectasia mutated protein (ATM), topoisomerase II, 70 kDa heat shock protein (HSP70), and p53 expression (Figure 4).

In contrast, enterolactone as a product isolated from berries has shown beneficial effects on colon cancer. A study by Danbara et al. [271] assessed the in vivo effects of enterolactone on colo 201 cells transplanted in athymic mice. The results showed that at a dose of 10 mg/kg administered 3 times per week by subcutaneous injection, enterolactone suppressed the growth of colo 201 cells (IC_50_ = 118.4 µM for 72 h) by modulating the apoptosis cascade and lowering the cell proliferative activity; in particular, an increase in the cleaved form of Caspase-3, Bcl-2, and downregulated proliferating cell nuclear antigen (PCNA) expression; while p53, Bax, Bcl-xL and S, and caspase-8 protein levels remained unchanged (Figure 4).

The chemopreventive activities of anthocyanin-rich extracts of grape (*Vitis vinifera*), bilberry (*Vaccinium myrtillus* L.), and chokeberry (*Aronia meloncarpa* E.) were investigated in rats previously treated with a colon carcinogen. The number and multiplicity of colonic aberrant crypt foci (ACF) and lesions used as biomarkers for colon cancer development decreased (*p* < 0.05) for the three berry extracts tested. Chokeberry and bilberry treatments were the most effective on these parameters. Nevertheless, COX-2 mRNA levels were downregulated by only grape and bilberry extracts. The inhibition of COX-2 expression led to reduced ACF in the colon, thereby promoting the chemopreventive effect observed for berries [272].

##### Clinical Trials

There have been a limited number of human studies investigating the effects of berries on colon cancer; to our knowledge, only black raspberry has been investigated. Wang et al. [273] showed that oral administration of 60 g per day of BRB powder for 1 to 9 weeks in 20 CRC patients was significantly effective for approximately 4 weeks in modulating the Wnt pathway via demethylation of three tumor-suppressor genes (SFRP2, SFRP5, and WIF1), as well as in decreasing the expression of DNA methyltransferase 1 (DNMT1); the components of BRBs that exerted demethylation effects were not elucidated. BRBs could also reduce the expression of genes associated with apoptosis (TUNEL), cell proliferation (Ki-67), angiogenesis (CD105), and the Wnt pathway (β-catenin, E-cadherin, and c-Myc) in adjacent normal tissues and colorectal tumors. Other work performed by Wang et al. [274] showed that BRB anthocyanins were responsible, at least in part, for the demethylation of DNMT1 and DNMT3B in cultured human CRCs. 

In another study, Pan et al. [275] focused on the metabolic pathways altered by freeze-dried BRBs in a group of 28 CRC patients treated with 60 g of BRB powder daily for a period of 1 to 9 weeks. The results showed changes in metabolites (alterations in amino acid and lipid metabolites, energy, and benzoates derived from BRB components) in both plasma and urine. An increased level of 4-methyl-catechol sulfate in plasma and urine after the intervention was significantly correlated with a higher level of apoptotic marker in tumors. BRBs should be beneficial to CRC patients through the regulation of multiple metabolites.

Plasma cytokines (IL-1β, IL-2, IL-6, IL-8, IL-10, IL-12p70, GM-CSF, IFN-γ, and TNF-α) can also be used as potential noninvasive indicators for monitoring tissue response to berry-based interventions for CRC. Treatment of 24 patients with CRC who consumed BRB powder (20 g in 100 mL drinking water) three times daily for 9 weeks showed a protective effect against CRC. Plasma cytokines were associated with berry treatment and changes in CRC tissue markers of apoptosis, cell proliferation, and angiogenesis. Particularly, the researchers observed an increase in plasma GM-CSF and a decrease in plasma IL-8 in patients receiving berries for more than 10 days [253]. It should be mentioned that high levels of IL-8 in the human tumor microenvironment are considered a limiting biomarker in cancer progression, invasion, and metastasis [276]. Consistent with these findings, in vitro studies showed that raspberries may also be involved in attenuated IL-8 secretion in cell culture [277].

Familial adenomatous polyposis (FAP) caused by mutations in the adenomatous polyposis coli (APC) gene can lead to colonic polyposis and a high risk of CRC. BRB suppositories (each containing 720 mg of BRB) had beneficial effects on the regression of rectal polyps in FAP patients treated daily with BRBs for 9 months. However, no increased benefit was observed when patients received the oral treatment in addition to the suppository. This study suggests that BRBs decreased cell proliferation, p16 promoter methylation, and DNMT1 protein expression, and regulated the Wnt pathway via miRNA demethylation in adenomas of FAP patients, but did not exert any effects on the promoter methylation of SFRP2 and WIF1 [278].

#### 5.2.2. Breast Cancer

Breast cancer has become the most common cancer and the second-leading cause of cancer death in women worldwide [64,279]. Several risk factors are involved in the increase in breast cancer, including diet, lifestyle, age, gender, overweight, cumulative exposure to estrogen, alcohol, radiation, family history, smoking, and genetic factors [280]. The gut microbiota may be influencing clinical outcomes and side effects of early breast cancer management [281]. In addition, breast and gut microbiota are key factors in promoting breast health [282]. 

##### In Vitro Studies

Cyanidin-3-glucoside (C3G), an anthocyanin found to be rich in edible fruits such as black raspberry and strawberry, plays a crucial role in inhibiting the progression of breast cancer through its antioxidant and anti-inflammatory properties. C3G treatment in a dose-dependent manner could repress breast-cancer-induced angiogenesis by inhibiting the expression and secretion of vascular endothelial growth factor (VEGF). In addition, C3G could attenuate transducer and activator of transcription 3 (STAT3) expression at both the protein level and in mRNA that transcriptionally activate VEGF [279]. Furthermore, Xu et al. [283] revealed that C3G blocked ethanol-induced migration/invasion of breast cancer cells overexpressing ErbB2 via phosphorylation inhibition of ErbB2, cSrc, FAK, and p130Cas pathway. Additionally, a study performed by Li et al. [284] confirmed that C3G treatment inhibited the overexpression of the human epidermal growth factor receptor 2 (HER2) tyrosine kinase receptor [285], induced apoptosis of trastuzumab-resistant breast cancer cells, and suppressed their migration and invasion. C3G isolated from mulberry also was shown to prevent cancer growth by inducing apoptosis on MDA-MB-453 human breast cancer cells by increasing the level of cleaved caspase-3 while decreasing Bcl-2 and DNA fragmentation [286]. Strawberry was able to induce cytotoxicity in T47D breast cancer cells in vitro [287]. 

##### In Vivo Studies

Some types of berry fruits have shown chemopreventive effects on breast cancer cells. Blueberry was investigated by Jeyabalan et al. [288] against 17β-estradiol (E2)-mediated mammary tumorigenesis in rats supplemented with a 5% blueberry diet, 2 weeks before or 12 weeks after E2 treatment in preventive and therapeutic modes. The tumor volume and multiplicity were reduced significantly in both intervention modes, while the tumor latency for palpable mammary tumors was delayed by 28 and 37 days. These data were supported by another experimental study carried out by Ravoori et al. [289] in which a small dose of blueberry diet (2.5%) given to August Copenhagen Irish (ACI) female rats with mammary tumors was sufficient to reduce tumor proliferation. The anticarcinogenic effects of blueberry on mammary tumorigenesis were largely due to the downregulation of CYP1A1 and ER-α gene expression and modulation of microRNA (miR-18a and miR-34c) levels, as well as the control of E2 metabolism and signaling [288,289]. Black raspberry and blueberry diets have shown chemopreventive and therapeutic properties against estrogen-mediated breast cancer. Black raspberry effectively delayed tumor latency and downregulated ERα expression, while blueberry showed more efficacy in reducing the proliferation of breast tissue. Tumor burden and downregulation of CYP1A1 expression due to the distinct phytochemicals [289], including anthocyanins, protocatechuic acid, quercetin, and ellagic acid, exhibited anticancer potentials [290].

In another study, Saarinen et al. [291] studied the effects of lariciresinol (a lignan found in some berries) in dimethylbenze(a)anthrazene (DMBA)-induced mammary cancer in rats treated with 3 or 15 mg/kg after 10 weeks. The results showed that there was no reduction in the multiplicity or incidence of these already-established tumors, and lariciresinol significantly inhibited tumor growth, surface area, and tumor angiogenesis after 9 weeks of administration.

On the other hand, methanolic extract of strawberry (MESB) can act as both a chemopreventive and therapeutic agent in mice bearing breast adenocarcinoma, which abrogates the proliferation of cancer cells by inducing apoptosis through the modulation of the expression of p73 and by activating the mitochondrial pathway of apoptosis [287]. Moreover, Jamun fruit extract obtained from ripe purple berries revealed antiproliferative and proapoptotic potential on MCF-7aro and MDA-MB-231 cells, but exhibited only mild antiproliferative efficacy and no proapoptotic effect on the normal MCF-10A breast cells [292].

Cranberry extract contained abundant phenolic compounds and possessed the highest level of antioxidant activity [293]. It had the ability to inhibit the proliferation of MCF-7 human breast cancer cells at doses of 5 to 30 mg/mL, and this suppression could be partly explained by the initiation of apoptosis after 4 h of treatment and the G_1_ phase arrest between 10 and 24 h of treatment. The mechanism of action involved in G_1_ arrest and apoptosis could be attributed to the upregulation of p21, as well as the downregulation of cyclin D1 and CdK4 [294].

##### Clinical Trials 

Although there is no direct evidence that berry consumption in adulthood prevents the onset of breast cancer, the mechanistic effect of berry constituents may support this hypothesis. Evidence from in vitro and in vivo studies indicated that berry polyphenols had an antiestrogenic effect in the presence of E2 [246]. Consumption of adult berries may have a protective effect in women at high risk, including late menopause, high circulating E2 levels before menopause, and first menstruation. Clinical trials can be conducted in high-risk women, and changes in circulating E2 levels, antioxidant status, plasma and urinary polyphenols/metabolites, and other relevant biomarkers can be assessed as indicators of risks and benefits [295].

#### 5.2.3. Esophageal Cancer

Esophageal cancer (EC) is one of the most aggressive malignant diseases reported and the leading cause of death. Berry phytochemicals for esophagus cancer were examined with preventive and therapeutic potentials to provide additional insights for future investigation on novel drug synthesis [296].

##### In Vitro Studies

A study by Chuang-Xin et al. [297] demonstrated in vitro that berry-derived quercetin was able to act synergistically with 5-FU, a chemotherapeutic agent for the treatment of EC, to inhibit growth (*p* < 0.05) and induce apoptosis (*p* < 0.005) in human esophageal cancer cells (EC9706 and Eca109) compared with the use of quercetin or 5-FU alone. These effects were attributed to the attenuation of NF-κB activation by the decrease in pIκBα expression. 

Moreover, gallic acid isolated from strawberries showed a significant inhibition of cell proliferation (IC_50_ = 0.2 mg/mL) and stimulated apoptosis in esophageal cancer cells (TE-2) in vitro. The molecular mechanism involved in apoptosis revealed that this acid downregulated the Akt/mTOR pathway and antiapoptosis proteins such as Bcl-2 and Xiap. In addition, gallic acid promoted the upregulation of the proapoptosis protein, Bax, and induced caspase-cascade activity in esophageal cancer cells [298].

##### In Vivo Studies

Through the similar molecular mechanisms of action of the berry components mentioned above, anthocyanins in BRB (5% BRB diet) and strawberries (5% and 10% dietary) have been found to contribute to the chemopreventive activity of N-nitroso-methyl benzylamine (NMBA) of induced esophageal cancer in rats [299,300].

Wang and colleagues [301] also investigated the chemopreventive effects of BRB in late-stage rat esophageal tumorigenesis. Animals were supplemented with 5% lyophilized BRB and evaluated 35 weeks after NMBA treatment. The results showed that BRBs reduced the number of dysplastic lesions, as well as the size of esophageal papillomas, in NMBA-treated rats, and contributed to the modulation of mRNA expression of genes associated with lipid and carbohydrate metabolism, cell proliferation and death, and inflammation. Furthermore, these findings suggested that anthocyanins and unknown compounds from BRBs modulated the expression of proteins associated with proliferation, apoptosis, and inflammation, as well as angiogenesis, and both cyclooxygenase and lipoxygenase pathways of arachidonic acid metabolism in NMBA-treated rat esophagus.

##### Clinical Trials 

In humans, Chen and colleagues [302] tested freeze-dried strawberry supplementation in patients with esophageal dysplastic lesions at high risk of developing esophageal cancer. During the experiment, subjects consumed either 60 or 30 g (n = 36 per group) of freeze-dried strawberry powder as a water-based drink for 6 months. At the end of treatment, subjects consuming the 60 g dose showed significantly lower protein expression of iNOS (79.50% reduction), COX-2 (62.9% reduction), pNF-κB-p65 (62.6% reduction), and pS6 (73.2% reduction) in human esophageal mucosa (Figure 5). Reductions in the same parameters were also noted in the group receiving 30 g of strawberry powder daily, but the changes were not large enough and/or required a longer intervention to have chemoprevention potentials.

It has been established that human esophageal squamous cell carcinoma is characterized by overexpression of COX-2 [303], iNOS [304], NF-κB [305], and mammalian target of rapamycin (mTOR) [306] (Figure 5). The findings of these studies should prove that berry polyphenols such as anthocyanin strawberries and BRBs inhibit these pathways, and may have chemopreventive effects in esophageal squamous cell carcinoma.

#### 5.2.4. Prostate Cancer

Prostate cancer remains the most common cancer and the second-leading cause of death in men [307]. The gut microbiota impacts disease progression, pathogenesis, and response to medical treatment in this genitourinary malignancy [308]. In fact, the commensal gut microbiota has been shown to play a role in endocrine resistance in castration-resistant prostate cancer (CRPC) by providing an alternative source of androgens [309]. 

Little is known about the effect of raspberries on granulocyte-macrophage colony-stimulating factor (GM-CSF), except that quercetin, a bioflavonoid component of raspberry, stimulated the production of GM-CSF and the recruitment of dendritic cells to human prostate cancer PC-3 cells following classical endoplasmic reticulum (ER)/Golgi pathways, thus enabling host immunosurveillance [310] and considering a novel approach to avoid the significant toxicity associated with the IFN and IL-2 based immunotherapy [311]. According to another study, berry extracts demonstrated their ability in vitro to inhibit the growth of the LNCaP prostate carcinoma cell line through antiproliferative activity [260].

Ellagic acid is a polyphenolic compound present in berries such as strawberries, raspberries, and blackberries, inducing apoptosis and growth inhibition of human prostate cancer PC3 cells in vitro. These findings suggested that the molecular basis of the apoptotic effect of ellagic acid may collectively lead to the cleavage of the PARP protein, the upregulation of Bax, and the downregulation of Bcl2 [312]. 

A study by Neto et al. [264] showed inhibition of matrix metalloproteinase expression in DU 145 prostate tumor cells by whole cranberry extract and proanthocyanidin fractions. Consequently, cranberry extract inhibited the expression of the matrix metalloproteinases MMP-2 and MMP-9 in the DU 145-cell line at a concentration of 100 µg/mL. The cranberry proanthocyanidin fraction (500 µg/mL) completely inhibited MMP-2 expression, and resulted in approximately 75% inhibition of MMP-9 activity. This finding could be attributed in part to proanthocyanidins, but it appeared that other cranberry components such as quercetin and ursolic acid contributed to the observed inhibition of MMPs. Other studies confirmed that an acidified methanol eluate (fraction 6, or Fr6) of cranberry containing flavonoids demonstrated antiproliferative activity, and had the greatest effect on LNCaP (IC_50_ = 9.9 mg/L of Fr6) among the several types of human tumor cell lines tested [265]. In addition, it was concluded that gut bacteria, functioning through short-chain fatty acids, regulated systemic and local insulin-like growth factor-1 (IGF1) in the host prostate, which may enhance the growth of prostate cancer cells in vivo [313].

Further human studies are needed to provide clear evidence of the protective effect of berry-extracted polyphenols on human health to better assess the benefit/risk ratio that may result from excessive consumption of polyphenols.

### 5.3. Beneficial Effects on Gut-Microbiota-Induced Metabolic Disorders 

Metabolic syndrome is defined as a pathologic condition characterized by a cluster of risk factors that often include abdominal obesity, dyslipidemia, glucose intolerance, insulin resistance, and hypertension, as well as a pro-oxidant, proinflammatory, and prothrombotic environment [7,314]. The two basic forces spreading this disorder are the overconsumption of energy-dense, fiber-containing fast food and a sedentary lifestyle, which are considered to be the major causes of abnormal glucose metabolism and obesity-associated insulin resistance [187]. This has led to a dramatic spread of disorders such as type 2 diabetes mellitus, high blood triglycerides, altered cholesterol levels, cardiovascular diseases, strokes, and other disabilities [314]. The metabolic activities of the gut microbiota ultimately result in the extraction of calories from ingested food substances [229]; therefore, evidence supports the importance of diet and its impact on the composition of the gut microbiota in animals and humans. A diet that is essentially high in lipids and low in fiber has been associated with the overgrowth of pathogenic flora, causing an intestinal dysbiosis that leads to increased endotoxemia, inducing the chronic inflammation involved in the pathophysiology of diseases, such as obesity and diabetes [240]. Based on scientific evidence, researchers have suggested that consuming berry fruits has a beneficial effect in the prevention and treatment of the majority of risk factors for metabolic disorders. This could be due to the presence of polyphenols with known antioxidant and anti-inflammatory properties, such as anthocyanins and/or phenolic acids [7,315].

#### 5.3.1. In Vivo Analysis

In order to elucidate the antiobesity potential, anti-inflammatory status, and lipid profile enhancement of berries, de Souza et al. [13] showed that dietary consumption of 2% açai pulp for 6 weeks improved oxidative stress biomarkers and lipid profiles in the serum of hypercholesterolemic rats treated with high-fat diets. Açai supplementation significantly reduced superoxide dismutase (SOD) activity and increased paraoxonase activity. Additionally, sweet cherry anthocyanins showed antiobesity properties in male C57BL/6 mice fed a high-fat diet for 12 weeks by reducing the adipocyte cell size and body weight gain, as well as the serum leptin secretion, serum glucose, triacylglycerol, total cholesterol, and LDL cholesterol, as well as reducing the expression levels of IL-6 and TNF-α genes and dramatically increasing the activity of antioxidant enzymes such as glutathione peroxidase (GPx) and SOD [316]. In addition, anthocyanin-rich phytochemicals in aronia fruits (chokeberry) suppressed visceral fat accumulation in rats fed a high-fat diet by inhibiting pancreatic lipase activity and intestinal lipid absorption [317]. Black currant, lingonberry, and blueberry supplementation were shown to decrease hepatic lipid accumulation, body fat content, and plasma levels of the inflammatory marker PAI-1 in high-fat-fed C57BL/6J mice [318].

According to another study, polyphenol-rich bilberry had the ability to reduce the levels of total cholesterol (−60%) and LDL cholesterol (−21%) in the diet of Zucker diabetic fatty rats, probably due to their high anthocyanin content, but HDL-cholesterol levels were increased slightly [319]. Moreover, Benn et al. [320] demonstrated that black currant improved the lipid profile in mice fed a high-fat/high-cholesterol diet supplemented with a black currant extract. Polyphenol-rich black currant reduced hypercholesterolemia, hyperglycemia, and liver steatosis, and also prevented diet-induced metabolic disturbances through the modulation of proprotein convertase subtilisin/kexin type 9 (PCSK9)/low-density lipoprotein receptor (LDL-R) pathways.

In addition, in Fischer rats, supplementation with anthocyanin-rich grape-bilberry juice (1551 mg anthocyanins/L) for 10 weeks decreased serum cholesterol, serum leptin, and resistin concentrations and ameliorated plasma fatty-acid composition, compared to animals supplemented with polyphenol-depleted grape–bilberry juice. These findings suggested that anthocyanins have a preventive potential for diseases associated with obesity, in particular cardiovascular diseases or type 2 diabetes [239].

In the aim to evaluate the antidiabetic potential of berries in vivo, Eid et al. [321] confirmed that lingonberry (*Vaccinium vitis-idaea* L.) represents a treatment option for diabetes, paving the way to clinical trials. The authors showed that treatment with an ethanol extract of *V. vitis-idaea* berries significantly decreased glycemia and insulin levels in obese and insulin-resistant mouse models. These findings were correlated with an increased translocation of glucose transporter GLUT4 content and activation of the AMPK and PI3-K/Akt pathways in skeletal muscle. Lingonberry treatment also enhanced hepatic steatosis by decreasing hepatic triglyceride levels and significantly activating liver AMPK and Akt pathways. In the same context, black chokeberry and elderberry have modulated immune system imbalances in insulin-deficiency diabetes. Polyphenolic extracts administrated to Wistar rats (0.040 g/kg b.w. every 2 days for 16 weeks) significantly reduced the production of proinflammatory cytokines, including TNF-α and IFN-γ, and self-sustained pancreatic insulitis [322].

Resveratrol, a polyphenol widely found in fruits such as blueberries, raspberries, and mulberries, has been studied in various animal models and human studies, and has shown beneficial effects in metabolic syndrome in terms of glucose and lipid homeostasis and reduction in body fat accumulation [323]. Resveratrol has been shown to mimic the effects of calorie restriction by activating protein deacetylase, sirtuin 1 (SIRT1), and peroxisome proliferator-activated receptor γ coactivator 1-α (PGC-1 α), and thus prolonged the lifespan of mice and prevented damage associated with excessive caloric intake in rodents, such as body fat accumulation and insulin resistance [324,325].

#### 5.3.2. In Vitro Studies

In vitro investigations by Lee and colleagues [326] provided evidence that anthocyanins influenced adipocyte function and lipogenesis pathways during adipocyte differentiation in 3T3-L1 cells by inhibiting lipogenic and adipogenic genes. Furthermore, anthocyanins markedly decreased protein and gene expression levels of key transcription factors such as liver X receptor α (LXRα), sterol regulatory element-binding protein-1c (SREBP-1c), peroxisome proliferator-activated receptor-γ (PPARγ), and CCAAT enhancer-binding protein-α (C/EBPα). Therefore, consumption of berry anthocyanins may play a pivotal role in preventing obesity and metabolic syndrome. Similarly, Takahashi et al. [317] demonstrated that aronia phytochemicals were effective in inhibiting increases in plasma triglyceride levels (IC_50_ = 1.50 ± 0.01 mg/mL) by inhibiting pancreatic lipase activity. In the same sense, Noguchi et al. [327] observed a reduction in serum triglyceride levels in boysenberry-fed rats, leading to the inhibition of gut fat absorption via boysenberry phytochemicals.

In order to show the antidiabetic effect of berries, Boath and colleagues [328] confirmed that polyphenols from strawberry, arctic bramble, lingonberry, cloudberry, and raspberry had the ability to inhibit both α-amylase and α-glucosidase activity in vitro. The inhibition was caused at low levels (cloudberry: IC_50_ = 5 μg GAE (gallic acid equivalents)/mL phenols), achievable in the gastrointestinal tract after ingestion of several berries, and approached a similar inhibition level compared to orlistat, a pharmaceutical lipase inhibitor [328].

Similarly, polyphenol-rich extracts from certain berries, notably black currant and rowanberry, have been identified as the most effective in modulating starch digestion by inhibiting α-glucosidase activity in vitro, with IC_50_ values of 20 and 30 μg GAE/mL, respectively. Interestingly, both black currant (rich in anthocyanins, ~70% of the total) and rowanberry (rich in chlorogenic acids, ~65% of the total) acted synergistically with acarbose, a pharmaceutical glucosidase inhibitor, and could substitute for acarbose or reduce the dose required for effective glycemic control in patients with type 2 diabetes [329].

Moreover, using multiple cellular bioassays, a study carried out by Martineau [330] proved that the lowbush blueberry *Vaccinium angustifolium* Ait. possessed insulin-like and glitazone-like activities, as well as cytoprotective activities. Furthermore, *V. angustifolium* exhibited potential antidiabetic effects in pancreatic β-cells, enhancing glucose transports in C_2_C_12_ muscle cells and 3T3-L1 adipocytes in the absence of insulin.

Berry anthocyanins are potent antioxidants and can influence glycemic control by protecting pancreatic β-cells from glucose-induced toxicity and oxidative stress, as well as inhibiting crucial enzymes involved in starch and lipid digestion, which gives them a crucial role in the prevention of metabolic diseases [331].

#### 5.3.3. Clinical Trials

Human studies revealed that berries have an emerging therapeutic role in reducing total cholesterol, LDL cholesterol, and triglyceride levels, and in increasing HDL cholesterol, and may thus enhance lipid status in patients with diabetes and hypertension [9]. Zhu et al. [332] performed a meta-analysis of randomized controlled trials in order to investigate the effect of the anthocyanin-rich berry *Vaccinium* on serum lipids. The results showed that whortleberry had a beneficial effect on lipid reduction, which was manifested by an increase in the levels of total cholesterol, triglycerides, and LDL-C, and at the same time, an increase in HDL-C. The molecular mechanism involved in the lipid-lowering potentials of anthocyanins has been attributed to the suppression of cholesteryl ester transfer protein (CETP) and the inhibition of LDL oxidation, as well as the enhancement of the activity of HDL-associated paraoxonase 1 [333].

In an attempt to re-emphasize the improvement by berries on the lipid profile, a randomized, double-blind, placebo-controlled trial of 120 dyslipidemic subjects aged between 40 and 65 years showed that consumption of 320 mg/day of purified blueberry and black currant anthocyanins for 12 weeks decreased LDL cholesterol and increases HDL cholesterol concentrations partially mediated through the inhibition of CETP. However, anthocyanin supplementation did not affect total cholesterol, triglyceride, apolipoprotein (apo) A-I, apo B, or glucose levels [334]. 

A study by Basu et al. [335] assessed the effects of blueberry supplementation on 48 obese men and women with metabolic syndrome. Subjects received a daily blueberry drink consisting of 50 g of freeze-dried blueberries for 8 weeks in a randomized controlled trial. The results showed significant decreases in systolic and diastolic blood pressure, plasma oxidized LDL, serum malondialdehyde, and 4-hydroxynonenal concentrations. No changes were reported for triglyceride and cholesterol levels; the inflammatory biomarkers CRP, IL-6, and adiponectin; or the adhesion molecules vascular cell adhesion molecule-1 (VCAM-1) and intercellular adhesion molecule-1 (ICAM-1) [335]. 

Regarding antioxidant potential, Riso et al. [336] proved that regular consumption of a wild blueberry drink consisting of 25 g of wild blueberry powder by 18 subjects with risk factors for cardiovascular disease for 6 weeks significantly reduced the levels of oxidized DNA bases and increased the resistance to oxidatively induced DNA damage in blood mononuclear cells, but no changes were noted in blood endothelial function, lipid profile, or inflammation markers [336].

On the other hand, Stull et al. [337] conducted a randomized, controlled clinical study to evaluate the impact of daily dietary supplementation with bioactive blueberry compounds (45 g of freeze-dried blueberry powder per day for 6 weeks) on 32 obese men and women resistant to insulin. At the end of the study, blueberries enhanced insulin sensitivity in obese, nondiabetic, and insulin-resistant participants, but there were no significant changes in adiposity, energy intake, or inflammatory biomarkers such as high sensitivity C-reactive protein (hsCRP), TNFα, and monocyte chemoattractant protein 1 (MCP-1) in the blueberry group compared to the control group [337].

In a randomized, placebo-controlled trial in 58 men with type 2 diabetes given 240 mL of cranberry juice daily for 12 weeks and in 56 healthy humans supplemented with 240 mL of low-calorie cranberry juice twice daily for 4 weeks, cranberry consumption improved lipid status by increasing serum HDL-C, serum apoA-1, and paraoxonase-1 (PON-1) activity, and by decreasing serum total cholesterol, triglycerides, serum glucose, and apoB, as well as diastolic blood pressure, insulin resistance, and CRP. Thus, cranberry treatment should contribute to preventing the high risks of dyslipidemia, diabetes, and cardiovascular disorders [338,339]. In contrast, the administration of six capsules daily of concentrated cranberry powder (equivalent to 240 mL cranberry juice) in a randomized placebo-controlled trial for 12 weeks to a group of 27 subjects with type 2 diabetes showed no significant effects on blood glucose, glycated hemoglobin (HbA_1c_), triglyceride, HDL, or LDL levels [340]. 

In another randomized, controlled trial, the consumption of freeze-dried strawberries by 58 patients with type 2 diabetes showed an improvement in certain cardiovascular risk factors by reducing total cholesterol concentrations and the total cholesterol to HDL-C ratio, but not triglyceride or HDL-C levels [341]. 

Moreover, in a randomized, clinical study in 20 healthy female volunteers, the effect of black currants and lingonberries was shown to optimize postprandial metabolic responses to sucrose, which may be explained by the delayed digestion of sucrose and the consequent slow absorption of glucose. The study also showed a slower rise in serum glucose and insulin and an improved glycemic profile [342]. Another randomized, placebo-controlled intervention performed in a group of 36 overweight subjects with type 2 diabetes supplemented with 50 g/day of freeze-dried strawberries for 6 weeks showed decreased concentrations of serum malondialdehyde (MDA), HbA_1c_, and hs-CRP, while markers of total serum antioxidant status were significantly increased, and blood glucose levels remained unaltered. Supplementation with freeze-dried berry products, as natural sources of antioxidants with a low glycemic index, could be considered as an adjunctive therapy to improve metabolic complications of type 2 diabetes [343].

The hypotensive effect of berries also was demonstrated by Tjelle and colleagues [344] in a randomized placebo-controlled trial in 134 healthy individuals. Systolic blood pressure was significantly reduced over time when consuming a commercially available berry juice made from red grapes, chokeberries, cherries, and bilberries for 12 weeks. The hypotensive effect was more pronounced in hypertensive than in normotensive people.

Additionally, within 6 weeks, daily consumption of whole freeze-dried blueberries increased natural killer cells and plasma redox capacity, and reduced blood pressure, augmentation index, central pulse wave velocity, and aortic systolic pressures in sedentary men and women [345]. 

Similarly, within 8 weeks, daily freeze-dried blueberry consumption resulted in significantly improved blood pressure and arterial stiffness in postmenopausal women with pre- and stage 1 hypertension. Both the systolic and diastolic blood pressure were significantly lower, which may have been due in part to the increase in NO production observed in the blueberry group [346,347]. 

## 6. Conclusions and Future Perspectives 

The number of studies concerning bioactive compounds in berries, especially phenolic acids, flavonoids, anthocyanins, and tannins, has increased greatly in recent years, and they have paid particular attention to the absorption and bioavailability of these compounds, and to a more determined confirmation of their benefits for human health. The current research results show that a large amount of these compounds reach the gut and are metabolized there into smaller molecules. Once in the gut, these polyphenols and their metabolites can modulate microorganism populations and prevent chronic inflammation, and consequently colon cancer and metabolic disorders. Bioactive compounds of berry-derived polyphenols exhibit these biological properties via different subcellular, cellular, and molecular mechanisms. However, the mechanisms remain to be fully explained, and studies should focus on other compounds in berries that could also be beneficial, either alone or in synergy with these compounds. On the other hand, plants that contain berry-derived polyphenols are used by people, and therefore can be considered as nutraceuticals against inflammation and related cancer. Importantly, using the pharmacological properties presented in this review related to berry bioactive compounds might promote the development of alternative approaches to berry polyphenols in the prevention and treatment of various diseases and disorders, and their use in the development of new drugs and improvements in existing treatment options will also be a very important issue in research priorities in the future. Further research is also needed regarding the issues related to genetic and biotechnological approaches in order to determine the best conditions to improve the yield and diversity of phenolic compounds in berries. 

## Figures and Tables

**Figure 1 molecules-27-03286-f001:**
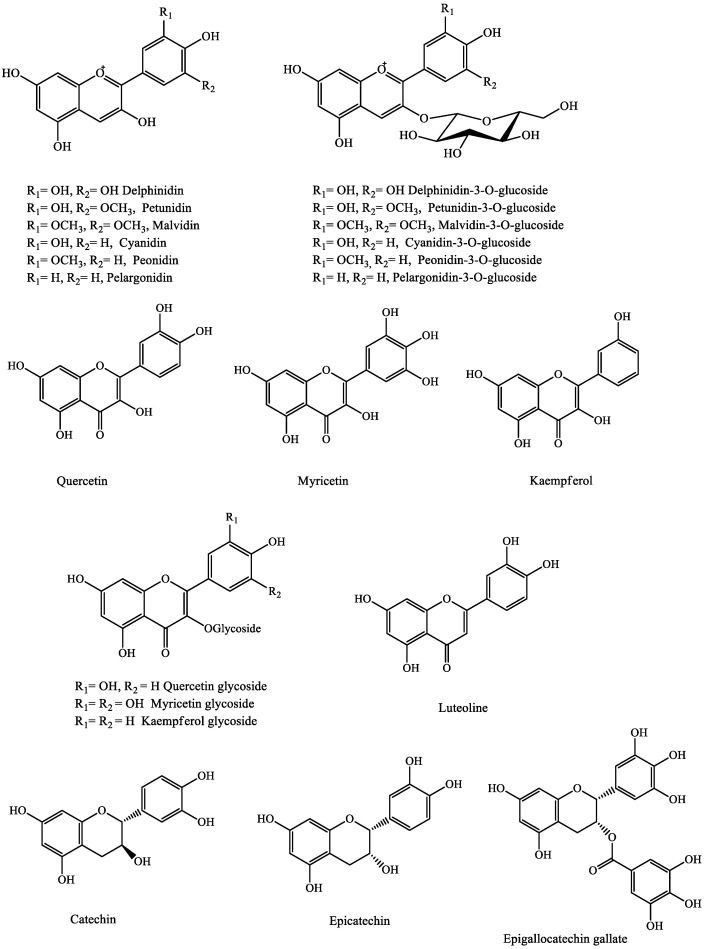
Chemical structures of major flavonoids contained in berries.

**Figure 2 molecules-27-03286-f002:**
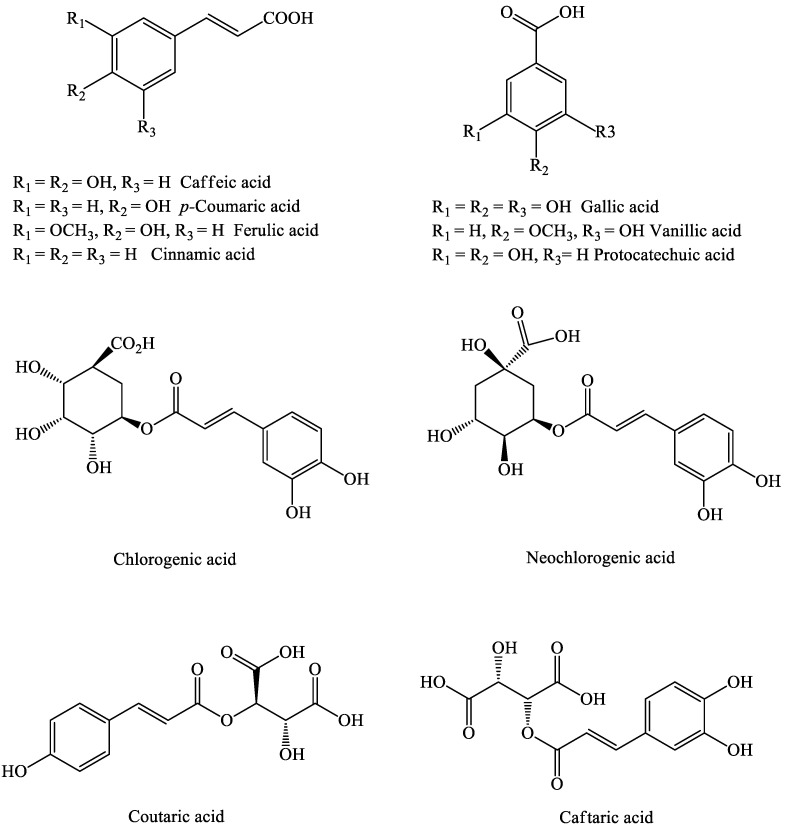
Chemical structures of phenolic acids contained in berries.

**Figure 3 molecules-27-03286-f003:**
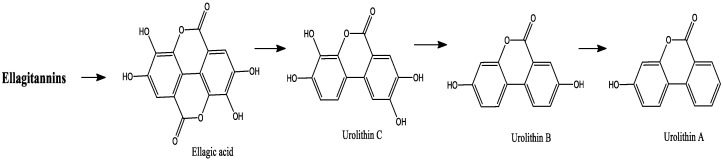
Microbial metabolism of ellagitannins.

**Figure 4 molecules-27-03286-f004:**
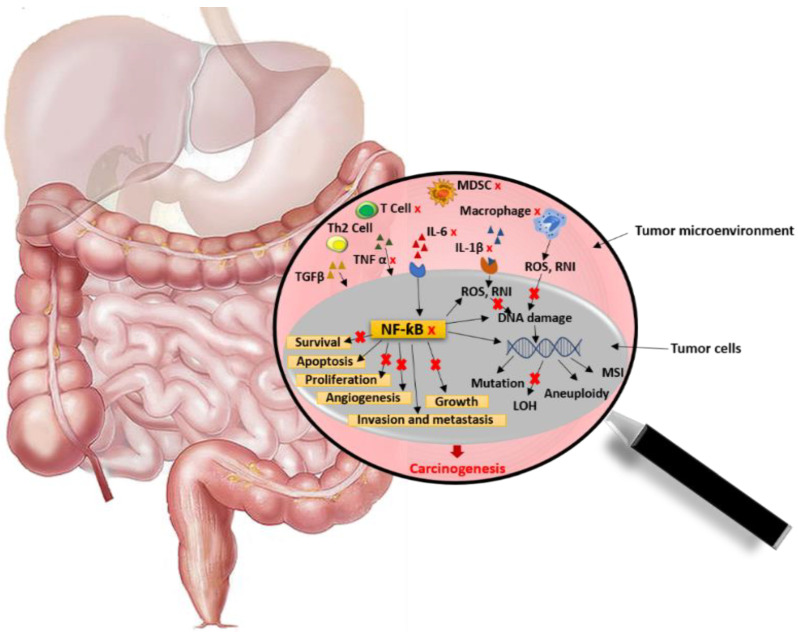
Mechanisms of CRC inhibition by berry polyphenols. MDSC: myeloid-derived suppressor cells; RNI: reactive nitrogen intermediates; LOH: loss of heterozygosity; MSI: microsatellite instability. Berry polyphenols target tumor cells by inhibiting NF-kB and related pathways, COX-2, iNOS, and the Bcl-2/Bax ratio. They affect the tumor microenvironment by inhibiting proinflammatory cytokine release (TNF-c, IL-1, IL-6 and IL-10, GM-CSF, and IL-8), T-cell proliferation and MDSC activity, and causing a decrease in macrophage and neutrophils infiltrated. Berry polyphenols also inhibit aneuploidy by reducing LOH and suppressing the expression of DNMT, DNMT, CDKN2A, SFRP2, SFRPS, and WIF1 in the Wnt pathway.

**Figure 5 molecules-27-03286-f005:**
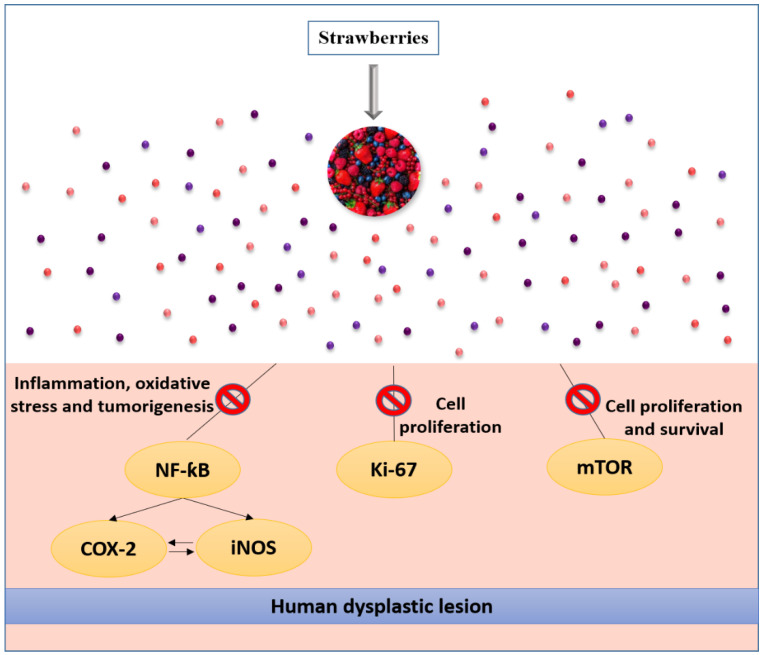
Inhibition mechanisms of esophageal precancerous progression by strawberries.

**Table 1 molecules-27-03286-t001:** Chemical compositions of berry polyphenols.

Plants	Characterization Methods	Major Polyphenols	References
*Vitis vinifera* cultivar: Mandilaria	HPLC–DAD	(+)-Catechin, (−)-epicatechin, procyanidin B2, and procyanidin B3	[88]
*Vitis vinifera* cultivar: Voidomatis	HPLC–DAD	(+)-Catechin, (−)-epicatechin, procyanidin B2, and procyanidin B3	
*Vitis vinifera* cultivar: Asyrtiko	HPLC–DAD	(+)-Catechin, (−)-epicatechin, procyanidin B3, and epicatechin gallate	
*Vitis vinifera* cultivar: Aidani	HPLC–DAD	(+)-Catechin, (−)-epicatechin, procyanidin B2, and procyanidin B3	
*Amelanchier alnifolia* Nutt. cultivar: Nelson	HPLC-ESI-MS/MS	Cyanidin-3-galactoside and chlorogenic acid	[89]
*Myrtus communis* from Canari	HPLC–DAD HS–SPME, GC and GC/MS LC–MS–MS	Myricetin, myricetin-3-*O*-arabinoside, and myricetin-3-*O*-galactoside	[90]
*Myrtus communis* from Bastia	HPLC–DAD HS–SPME, GC and GC/MS LC–MS–MS	Myricetin, myricetin-3-*O*-arabinoside, and myricetin-3-*O*-galactoside	
*Myrtus communis* from Agriate	HPLC–DAD HS–SPME, GC and GC/MS LC–MS–MS	Myricetin, myricetin-3-*O*-arabinoside, and myricetin-3-*O*-galactoside	
*Myrtus communis* from Corte	HPLC–DAD HS–SPME, GC and GC/MS LC–MS–MS	Myricetin, myricetin-3-*O*-arabinoside, and (−)epigallocatechin	
*Myrtus communis* from Ajaccio	HPLC–DAD HS–SPME, GC and GC/MS LC–MS–MS	Myricetin, quercetin-3-*O*-rhamnoside, and (−)epigallocatechin	
*Myrtus communis* from Morta	HPLC–DAD HS–SPME, GC and GC/MS LC–MS–MS	Myricetin, quercetin-3-*O*-rhamnoside, and (−)epigallocatechin	
*Myrtus communis* from Bonifacio	HPLC–DAD HS–SPME, GC and GC/MS LC–MS–MS	Myricetin, quercetin-3-*O*-glucoside, and myricetin-3-*O*-arabinoside	
*Myrtus communis* from Abbazzia	HPLC–DAD HS–SPME, GC and GC/MS LC–MS–MS	Myricetin-3-*O*-arabinoside, myricetin, and (−)epigallocatechin	
*Myrtus communis* from Travo	HPLC–DAD HS–SPME, GC and GC/MS LC–MS–MS	(−)Epigallocatechin, quercetin-3-*O*-rhamnoside, and myricetin	
*Myrtus communis* from Aleria	HPLC–DAD HS–SPME, GC and GC/MS LC–MS–MS	Myricetin, myricetin-3-*O*-arabinoside, and (−)epigallocatechin	
*Myrtus communis* L.	HPLC–DAD	Myricetin-3-*O*-arabinoside and myricetin-3-*O*-galactoside	[91]
*Hippophaërhamnoides* L.	HPLC	Catechin, epicatechin, gallic acid, *p*-coumaric acid, caffeic acid, ferulic acid, rutin (quercetin 3-rutinoside), and quercitrin (quercetin 3-rhamnoside)	[92]
Wild bilberry and blackberry cv. Čačanskabestrnaand cv. Thornfree	HPLC	Protocatechuic acid and gallic acid	[93]
*Vitis vinifera* cv. *Italia* (white)	LC/ESI-MS	Procyanidin B1, quercetin glucoronide, and epicatechin	[94]
*Vitis vinifera* cv. *Michele Palieri* (red)	LC/ESI-MS	Peonidin-3-*O*-glucoside, procyanidin B1, quercetin glucoronide, *cis*-resveratrol, and epicatechin	
*Vitis vinifera* cv. *Red Globe* (red)	LC/ESI-MS	Peonidin-3-*O*-glucoside, procyanidin B1, quercetin glucoside, *cis*-resveratrol, and epicatechin	
*Vaccinium* spp.	CIELAB and HPLC-DAD	Delphinidin-3- O-galactoside, malvidin-3-*O*-galactoside, malvidin-3-*O*-arabinoside, and delphinidin-3-*O*-arabinoside	[30]
*Vitis vinifera* cv. Aglianico	HPLC coupled with LC-ESI/MS/MS	Malvidin-3-(6-*O*-coumaroyl)-glucoside, malvidin-3-*O*-glucoside, *trans*-resveratrol, and petunidin-3-*O*-glucoside	[95]
*Lonicera caerulea* L.	UPLC-ESI-MS	Cyanidin-3-*O*-glucoside, quercetin-3-*O*-rutinoside, and catechin	[96]
*Vaccinium angustifolium*	HPLC and MS	Cyanidin-3-galactoside, delphinidin 3-glucoside, malvidin-3-galactoside, and petunidin-3-galactoside	[97]
*Aristotelia chilensis* (Molina) Stuntz	HPLC	Delphinidin-3-glucoside, delphinidin-3,5-diglucoside, delphinidin-3-sambubioside, and ellagic acid	[24]
*Vaccinium myrtillus*	HPLC	Gallic acid, vanillic acid, ferulic acid, and quercetin	[46]
*Vaccinium vitis-idaea*	HPLC	Vanillic acid, ferulic acid, caffeic acid, and *p*-coumaric acid	
*Rubus chamaemorus*	HPLC	Gallic acid, caffeic acid, ferulic acid, and *p*-coumaric acid	
*Hippophae rhamnoides* L.	HPLC	Quercetin, gallic acid, *p*-coumaric acid, and isorhamnetin	
*Rubus laciniatus* and *Rubus* sp. Hyb	HPLC	Epicatechin, ellagic acid, and rutin	[98]
16 strawberry cultivars	LC/DAD/ESI-MS	Ellagic acid, feruloyl, caffeoyl, coumaroyl hexose, quercetin, kaempferol, pelargonidin-3-*O*-glucoside, (epi)catechin, and (epi)afzelechin	[32]
*Ribes nigrum* L.	HPLC	Delphinidin-3-*O*-glucoside, delphinidin-3-*O*-rutinoside, cyanidin-3-*O*-glucoside, and cyanidin-3-*O*-rutinoside	[99]
*Vitis vinifera* L. cv. Shiraz	HPLC	(+)-Catechin, (–)-epicatechin, (–)-epicatechin-3-*O*-gallate	[100]
*Lonicera caerulea* L.	HPLC-DAD-MS	Cyanidin-3-glucoside	[101]
*Amelanchier alnifolia* Nutt.	HPLC-DAD and HPLC- ESI/MS	Chlorogenic acid, cinnamic acid deriv 2, cyanidin 3-galactoside, quercetin 3-galactoside, catechin deriv 1, and procyanidin deriv 2	[102]
*Prunus cerasus* cv. Marasca	HPLC	(–)-Epicatechin, quercetin 3-rutinoside, and chlorogenic acid	[31]
*Prunus cerasus* cv. Oblačinska	HPLC	(–)-Epicatechin, quercetin 3-rutinoside, and chlorogenic acid	
*Fragaria*× *ananassa* cv. Maya	HPLC	(–)-Epicatechin, kaempferol derivative, chlorogenic acid, and ellagic acid	
*Rubus idaeus* cv. Willamette	HPLC	(–)-Epicatechin, quercetin 3-rutinoside, chlorogenic acid, and ellagic acid	
*Vaccinium myrtilus*	HPLC	(–)-Epicatechin, kaempferol derivative, chlorogenic acid, and ellagic acid	
*Vitis labrusca*, *Vitis vinifera*, hybrid of *Vitis labrusca* and *Vitis vinifera*, hybrid of *Vitis vinifera* and *Vitis amurensis*, and hybrid of *Vitis thunbergii* and *Vitis vinifera*	HPLC and HPLC-MS	Delphinidin, cyanidin, petunidin, peonidin, malvidin, caftaric acid, coutaric acid, procyanindin B1, epicatechin, rutin, myricetin 3-*O*-glucoside, quercetin 3-*O*-glucuronide, and quercetin 3-*O*- glucoside	[48]
*Ribes nigrum* cv. Öjebyn, *Ribes nigrum* cv. Vertti, *Ribes pallidum* cv. Red Dutch, and *Ribes* × *pallidum* cv. White Dutch	LC-DAD and LC-MS	Caffeoylglucose, p-coumaroylglucose and hexose, feruloylhexose, myricetin 3-*O*-rutinoside, myricetin 3-*O*-glucoside, rutin, quercetin 3-*O*-glucoside, kaempferol hexoside−malonate, delphinidin 3-*O*-rutinoside, and cyanidin 3-*O*-rutinoside	[34]
Strawberry	HPLC-DAD	Pelargonidin 3-glucoside and pelargonidin 3-rutinoside	[25]
Blackberry	HPLC-DAD	Delphinidin 3-glucoside and cyanidin 3-glucoside	
Blueberry	HPLC-DAD	Petunidin 3-glucoside and delphinidin 3-glucoside	
Raspberry	HPLC-DAD	Cyanidin 3-glucoside and petunidin 3-glucoside	
*Gaultheria phillyreifolia* and *Gaultheria poeppigii*	HPLC-DAD-ESI-MS	Cyanidin galactoside, cyanidin arabinoside, delphinidin galactoside, delphinidin arabinoside, quercetin-3-arabinoside, quercetin-3-rutinoside, quercetin-3-rhamnoside and 3-caffeoylquinic acid, monotropein-10-*trans*-coumarate, monotropein-10-*trans*-cinnamate, and 6α-hydroxy-dihydromonotropein-10-*trans*-cinnamate	[35]
Blueberry, bilberry, cranberry, lingonberry, eastern shadbush, Japanese wineberry, black mulberry, chokeberry, red, black and white currants, jostaberry, red and white gooseberry, goji berry, and rowan; and wild and cultivated species of blackberry, raspberry, strawberry, and elderberry	HPLC–MS	Glycosides of quercetin, myricetin, kaempferol, isorhamnetin, syringetin, and laricitrin	[33]
*Myrtus communis* L.	HPLC-ESI-MS and HPLC-UV/VIS	Delphinidin-3-*O*-glucoside, cyanidin-3-*O*-glucoside, petunidin-3-*O*-glucoside, peonidin-3-*O*-glucoside, malvidin-3-*O*-glucoside, delphinidin-3-*O*-arabinoside, petunidin-3-*O*-arabinoside, malvidin-3-*O*-arabinoside, myricetin-3-*O*-galactoside, myricetin-3-*O*-rhamnoside, myricetin-3-*O*-arabinoside, quercetin-3-*O*-glucoside, quercetin-3-*O*-rhamnoside, and myricetin	[103]
White grapes cv. Rebula	HPLC-DAD	*Trans*-caftaric, *trans*-coutaric, *cis*-coutaric, and *trans*-fertaric acid	[104]
*Vitis vinifera* (Primitivo)	HPLC-PDA-ESI-MS	Catechin, petunidin-3-*O*-glucoside, peonidin 3-*O*-glucoside, and malvidin 3-*O*-glucoside	[105]
*Vitis vinifera* (Negroamaro)	HPLC-PDA-ESI-MS	Caftaric acid, delphinidin 3-*O*-glucoside, petunidin-3-*O*-glucoside, and malvidin 3-*O*-glucoside	
*Vitis vinifera* (Susumaniello)	HPLC-PDA-ESI-MS	Caftaricacid, catechin, delphinidin 3-*O*-glucoside, cyanidin 3-*O*-glucoside, petunidin-3-*O*-glucoside, peonidin 3-*O*-glucoside, *trans*-piceid, and malvidin 3-*O*-glucoside	
*Vitis vinifera* (Uva di Troia)	HPLC-PDA-ESI-MS	Catechin, delphinidin 3-*O*-glucoside, cyanidin 3-*O*-glucoside, petunidin-3-*O*-glucoside, peonidin 3-*O*-glucoside, and malvidin 3-*O*-glucoside	
*Vitis vinifera* (Malvasia Nera)	HPLC-PDA-ESI-MS	Catechin, epicatechin, petunidin-3-*O*-glucoside, peonidin 3-*O*-glucoside, and malvidin 3-*O*-glucoside	
*Vitis vinifera* (Aglianico)	HPLC-PDA-ESI-MS	Catechin, epicatechin, malvidin 3-*O*-glucoside, petunidin-3-*O*-glucoside, delphinidin 3-*O*-glucoside, and peonidin 3-*O*-glucoside	
*Vitis vinifera* (Cesanese)	HPLC-PDA-ESI-MS	Delphinidin 3-*O*-glucoside, cyanidin 3-*O*-glucoside, petunidin-3-*O*-glucoside, peonidin 3-*O*-glucoside, and malvidin 3-*O*-glucoside	
*Vitis vinifera* (Merlot)	HPLC-PDA-ESI-MS	Caftaric acid, epicatechin, delphinidin 3-*O*-glucoside, malvidin 3-*O*-glucoside, and catechin	
Red Globe grape (*Vitis vinifera* L.)	HPLC-ESI-QTOF-MS	Catechin, pelargonidin, and cyanidin-3,5-*O*-diglucoside	[26]
Raspberry (*Rubus idaeus* L.)	HPLC-ESI-QTOF-MS	Epicatechin, cyanidin-3,5-*O*-diglucoside, and pelargonidin	
Blackberry (*Rubus* spp.)	HPLC-ESI-QTOF-MS	Epicatechin, cyanidin-3-*O*-β-glucoside, and cyanidin-3,5-*O*-diglucoside	
Black currant (*Ribes nigrum*) cultivar ‘Titania’	HPLC-DAD and LC-ESI/MS	Chlorogenic acid, quercetin-3-*O*-galactoside, and quercetin-3-*O*-glucosyl-6’’-acetate	[106]
Raspberry (*Rubus ideaus*) cultivar ‘Polka’	HPLC-DAD and LC-ESI/MS	Chlorogenic acid, quercetin-3-*O*-rutinoside, and quercetin-3-*O*-glucuronide	
Honeysuckle (*Lonicera kamtschatica*) cultivar ‘Zielona’	HPLC-DAD and LC-ESI/MS	Chlorogenic acid, quercetin-3-*O*-glucoside, and 3,5-di-*O*-caffeoylquinic acid	
Bilberry (*Vaccinium myrtillus* L.)	HPLC-DAD and LC-ESI/MS	Chlorogenic acid, 3-*O*-*p*-coumaroylquinic acid, and quercetin-3-*O*-glucuronide	
Strawberry (*Fragaria* × *ananassa* Duch.) cultivar ‘Elkat’	HPLC-DAD and LC-ESI/MS	Quercetin-3-*O*-rutinoside, quercetin-3-*O*-glucuronide, and kaempferol-3-*O*-rutinoside	
*Vaccinium floribundum* Kunth	HPLC-UV/DAD HPLC-ESI-MS and MS	Chlorogenic acid, quercetin-3-*O*-galactoside, quercetin-3-*O*-arabinofuranoside, quercetin-3-*O*-rhamnoside, cyanidin-3-*O*-galactoside, cyanidin-3-*O*-arabinoside, and delphinidin-3-*O*-arabinoside	[29]
*Vaccinium myrtillus* L.	HPLC-UV/DAD HPLC-ESI-MS and MS	Chlorogenic acid, quercetin-3-*O*-galactoside, quercetin-3-*O*-glucuronide, delphinidin-3-*O*-galactoside, delphinidin-3-*O*-glucoside, cyanidin-3-*O*-galactoside, and petunidin-3-*O*-glucoside	
Wild blackberry (*Rubus fruticosus*)	HPLC	Gallic acid, *t*-caftaric acid, caffeic acid, and sirginic acid	[49]
Blackthorn (*Prunus spinosa* L.)	HPLC	Gallic acid, *t*-caftaric acid, *t*-coutaric acid, caffeic acid, and sirginic acid	
European cornel (*Cornus mas*)	HPLC	Gallic acid, *t*-caftaric acid, caffeic acid, *p*-coumaric acid, and sirginic acid	
*Lycium barbarum* L.	UHPLC-ESI-QTOF-MS	Cyanidin, ferulic acid, catechin, and luteolin	[107]
Calafate (*Berberis microphylla*)	HPLC-DAD-MS/MS	Delphinidin-3-glucoside, cyanidin-3-glucoside, petunidin-3-glucoside, peonidin-3-glucoside, malvidin-3-glucoside, delphinidin-3-rutinoside, cyanidin-3-rutinoside, petunidin-3-rutinoside, peonidin-3-rutinoside, malvidin-3-rutinoside, delphinidin-3,5-dihexoside, cyanidin-3,5-dihexoside, petunidin-3,5-dihexoside, peonidin-3,5-dihexoside, malvidin-3,5-dihexoside, and petunidin-3-rutinoside-5-glucoside	[108]
Maqui (*Aristotelia chilensis*)	HPLC-DAD-MS/MS	Delphinidin-3-glucoside, cyanidin-3-glucoside, delphinidin-3-sambubioside-5-glucoside, cyanidin-3-sambubioside-5-glucoside, delphinidin-3-sambubioside, cyanidin-3-sambubioside, delphinidin-3,5-diglucoside, and cyanidin-3,5-diglucoside	
Murtilla (*Ugni molinae*)	HPLC-DAD-MS/MS	Cyanidin-3-glucoside and peonidin-3-glucoside	
*Berberis microphylla*	HPLC-DAD-ESI-MS/MS	Delphinidin-3-glucoside, petunidin-3-glucoside, malvidin-3-glucoside, and petunidin-3.5-dihexoside	[109]
*Berberis ilicifolia*	HPLC-DAD-ESI-MS/MS	Delphinidin-3-glucoside, petunidin-3-glucoside, malvidin-3-glucoside, and cyanidin-3-glucoside	
*Berberis empetrifolia*	HPLC-DAD-ESI-MS/MS	Delphinidin-3-glucoside, petunidin-3-glucoside, malvidin-3-glucoside, and petunidin-3-rutinoside	
*Ribes magellanicum*	HPLC-DAD-ESI-MS/MS	Delphinidin-3-glucoside, delphinidin-3-rutinoside, cyanidin-3-glucoside, and cyanidin-3-rutinoside	
*Ribes cucullatum*	HPLC-DAD-ESI-MS/MS	Delphinidin-3-glucoside, delphinidin-3-rutinoside, cyanidin-3-glucoside, and cyanidin-3-rutinoside	
*Gaultheria mucronate*	HPLC-DAD-ESI-MS/MS	Cyanidin-pentoside and delphinidin-pentoside + cyanidin-3-galactoside	
*Gaultheria antarctica*	HPLC-DAD-ESI-MS/MS	Cyanidin-pentoside, delphinidin-3-galactoside, and cyanidin-3-galactoside + delphinidin-pentoside	
*Rubus geoides*	HPLC-DAD-ESI-MS/MS	Cyanidin-3-sambubioside	
*Myrteola nummularia*	HPLC-DAD-ESI-MS/MS	Delphinidin-3-glucoside and cyanidin-3-glucoside	
*Fuchsia magellanica*	HPLC-DAD-ESI-MS/MS	Cyanidin-3-glucoside	
Greek grape (*Vitis vinifera*) samples	HPLC	(+)-Catechin, (–)-epicatechin, *trans*-resveratrol, quercetin, and quercetin glycosides	[110]
*Luma apiculata*	HPLC-DAD-ESI/MS	Delphinidin 3-*O*-hexose, epigallocatechin gallate, cyanidin-3-*O*-glucose, petunidin 3-*O*-glucose, peonidin 3-*O*-glucose, malvidin 3-*O*-glucose, isoquercitrin, and quercitrin	[111]
*Luma chequén*	HPLC-DAD-ESI/MS	Procyanidin B1, delphinidin 3-*O*-hexose, cyanidin-3-*O*-glucose, petunidin 3-*O*-glucose, peonidin 3-*O*-glucose, malvidin 3-*O*-glucose, isoquercitrin, and syringetin-3-*O*-glucose	
*Vitis vinifera* L. ssp. *Sativa*	HPLC-DAD	Chlorogenic acid, caffeic acid, gallic acid, vanillin, feluric acid, ellagic acid, and *E*-resveratrol	[47]
Grape varieties (cabernet sauvignon, syrah, malbec, and merlot)	HPLC and UV	Quercetin and *trans*-resveratrol	[112]
Black elderberry (*Sambucus nigra* L.)	HPLC-DAD	Anthocyanins: cyanidin 3-sambubioside-5-glucoside, cyanidin 3,5-diglucoside, cyanidin 3-sambubioside, cyanidin 3-glucoside, and cyanidin 3-rutinoside Quercetins:quercetin, quercetin 3-rutinoside, and quercetin 3-glucoside	[28]
*Vaccinium uliginosum* berry	HPLC-DAD and HPLC-ESI-MS	Delphinidin 3-*O*-glucoside, malvidin 3-*O*-glucoside, myricetin 3-*O*-hexoside, and quercetin 3-*O*-galactoside	[113]
*Lonicera caerulea* berry	HPLC–DAD–EIS–MS	Cyanidin-3,5-dihexoside, cyanidin-3-hexoside-catechin, dimer of cyanidin-hexoside, peonidin-3,5-dihexoside, cyanidin- 3-glucoside, cyanidin-3-rutinoside, peonidin- 3-glucoside, peonidin-3-rutinoside, and dimer of cyanidin-3-hexoside	[61]
Black raspberry and marionberry	HPLC-DAD-ESI/MS/MS	Delphinidin, cyanidin, pelargonidin, petunidin, peonidin, and malvidin,	[23]
Blueberries (*Vaccinium uliginosum* L.)	HPLC-FT-ICR MS/MS	Cyanidin-3-*O*-glucoside, cyanidin-3-*O*-rutinoside, catechin, and myricetin 3-*O*-hexose	[114]
Bilberries (*Vaccinium myrtillus* L.)	HPLC-FT-ICR MS/MS	Cyanidin-3-*O*-rutinoside, catechin, quercetin-3-*O*-galactoside, and quercetin-3-*O*-arabinoside	
Mulberries (*Morus alba* L.)	HPLC-FT-ICR MS/MS	Malvidin-3-*O*-glucoside, myricetin 3-*O*-hexose, quercetin-3-*O*-rutinoside, and chlorogenic acid	
Cranberries (*Vaccinium oxycoccos* L.)	HPLC-FT-ICR MS/MS	Cyanidin-3-*O*-rutinoside, catechin, quercetin-3-*O*-rutinoside, and chlorogenic acid	
*Solanum scabrum*	HPLC/UV–visible/MS	Chlorogenic acid, neochlorogenic acid, quercetin, isorhamnetin, quercetin-rhamnosyl-hexoside, tigogenin, diosgenin, petunidin, delphinidin, malvidin, and petunidin-3-(*p*-coumaroyl-rutinoside)-5-*O*-glucoside	[115]
Black mulberry (*Morus nigra* L.)	HPLC-DAD-ESI HRMS	Quercetin-dirhamnosyl-hexoside, quercetin-rhamnosyl-dihexoside, and procyanidin trimer 1	[116]
Cornelian cherry (*Cornus mas* L.)	HPLC-DAD-ESI HRMS	Rhamnetin-rhamnosyl-hexoside, kaempferol-dirhamnosyl-hexoside, and procyanidin trimer (2 and 3)	
Elderberry (*Sambucus nigra* L.)	HPLC-DAD-ESI HRMS	Rhamnetin-rhamnosyl-hexoside, myricetin-rhamnosyl-hexoside, quercetin-acetyl-pentosyl-hexoside, kaempferol-rhamnosyl-dihexoside, quercetin-rhamnosyl-dihexoside, and procyanidin trimer 1	
Hawthorn (*Crataegus monogyna* L.)	HPLC-DAD-ESI HRMS	Rhamnetin-rhamnosyl-hexoside, quercetin-dihexoside, and procyanidin trimer (2 and 3)	
Lingonberry (*Vaccinium vitis-idaea* L.)	HPLC-DAD-ESI HRMS	Rhamnetin-rhamnosyl-hexoside and quercetin-dirhamnosyl-hexoside	
Rose hip (*Rosa canina* L.)	HPLC-DAD-ESI HRMS	Quercetin-galloyl-hexoside and procyanidin trimer 1	
*Rubus fruticosus*, *Ribes nigrum*, *Vaccinium corymbosum*, *Lyciumbarbarum* L., *Rubus idaeus*, *Ribes rubrum*, *Ribes grossularia* L., *Ribes pallidum*, and *Ribes grossularia* L.	HPLC-DAD-UV-ESI HRMS	Ferulic acid, coumaric acid, cyanidin, and delphinidin Myricetin, kaempferol, catechin, quercetin, chlorogenic acid, malvidin, and petunidin	[38]

**Table 2 molecules-27-03286-t002:** Summary of reports concerning the modulation of gut microbiota by berry polyphenols.

Berries Polyphenols	Type of Study	Dosage	Gut Microbiota Modulation	Ref.
Açai	In vitro	1 g of dry berries reduced in powder (24 h)	Inhibited the growth of *Clostridium histolyticum*, *Bacteroides-Prevotella* spp. *Bifidobacterium* spp., and *Lactobacillus*/*Enterococcus* spp.; *Clostridium coccoides-Eubacterium rectale* were not affected	[201]
Bilberry, black currant, buckthorn berry cloudberry, lingonberry, raspberry, and strawberry	In vitro	2 and 10 mg/mL of dry berries reduced in powder (24 h)	Inhibited the growth of *Salmonella enterica* and *Staphylococcus aureus* *Lactobacillus rhamnosus* and *Listeria monocytogenes* were not affected	[10]
Blackberry, black raspberry, black currant, maqui berry, Concord grape, and blueberry	In vivo (mice; high-fat diet)	400 μg berry anthocyanins (12 weeks)	Reduced gut luminal oxygenation and promoted abundance of obligate anaerobic bacteria from *Bacteroidetes* and *Actinobacteria* phyla at these sites	[187]
Black raspberry	In vivo (healthy mice)	10% of dry berry powder (6 weeks)	Increased the abundance of *Barnesiella* Reduced *Clostridium* and *Lactobacillus*	[140]
Blackberry	In vivo (rats; normal diet, and high-fat diet)	25 mg/kg/day (17 weeks)	Normal diet: promoted abundance of *Pseudoflavonifractor* and *Oscillobacter* High-fat diet: promoted abundance of *Oscillobacter* and reduced *Rumminococcus*	[208]
Black currant	In vivo (healthy rats)	2 mL (extracts; 4 weeks)	Promoted abundance of *Bifidobacteria* and *lactobacilli* Reduced *Bacteroides* and *clostridia*	[196]
Black currant	Clinical study (healthy volunteers)	672 mg/day of dry berry powder (2 weeks)	Promoted abundance of *Lactobacillus* spp. and *Bifidobacterium* spp. Reduced *Clostriudium* spp. and *Bacterioides* spp.	[198]
Black raspberry	In vivo (healthy rats)	5% of dry berry powder (6 weeks)	Promoted abundance of *Akkermansia*, *Desulfovibrio*, and *Anaerostipes*	[209]
Blueberry polyphenols			Altered gut bacteria composition, such as the abundance of *Proteobacteria*, *Deferribacteres*, *Actinobacteria*, *Bifidobacterium*, *Desulfovibrio*, *Adlercreutzia*, *Helicobacter*, *Flexispira*, and *Prevotella*	[184]
Blueberry	In vivo (mice; inflammatory bowel disease model)	10% of dry berry powder (21 weeks)	Reduced *Clostridium perfringens*, *Enterococcus* spp., *Lactobacillus* spp., and *Escherichia coli*	[210]
Blueberry	In vivo (healthy rats)	8% of dry berry powder (21 weeks)	Reduced *Lactobacillus* and *Enterococcus* Promoted abundance of bacteria from the orders *Actinomycetales*, *Bifidobacteriales*, and *Coriobacteriales*	[211]
Blueberry	In vitro	10 and 25% (berry extracts) for 5 days	Promoted abundance of *Lactobacillus rhamnosus* and *Bifidobacterium breve*	[212]
Blueberry	In vitro	5, 10, and 25% (berry extracts) for 48 h	Promoted abundance of *Lactobacilli* and *Bifidobacteria* populations	
Blueberry	Animal study (healthy rats)	4 mL berry extracts per kg per day for 6 days	Promoted abundance of *Lactobacilli* and *Bifidobacteria* populations	[212]
Blueberry	Human study (healthy volunteers)	250 mL drink (10% dry berry powder in water) for 6 weeks	Promoted abundance of *Bifidobacterium* spp. and *Lactobacillus acidophilus* The consumption of blueberry led to an increase in *Bifidobacterium* spp. and *Lactobacillus acidophilus*	[197]
Blueberry	Human study (healthy volunteers)	250 mL drink (10% dry berry powder in water) for 6 weeks	Promoted abundance of *Bifidobacterium longum* subsp. *infantis*	[213]
Cranberry	In vitro	(−) *Salmonella enterica*, *Staphylococcus aureus,* *Lactobacillus rhamnosus,* and (−) *Listeria monocytogenes*	Inhibited the growth of *Salmonella enterica*, *Staphylococcus aureus*, and *Listeria monocytogenes* *Lactobacillus rhamnosus* was not affected	[10]
Cranberry and grapeseeds	In vitro	500 mg/L of the extracts (48 h)	Inhibited the growth of *Bacteroides*, *Prevotella*, *Blautiacoccoides-Eubacterium* rectale, *Lactobacillus*, *Bifidobacterium*, *Enterobacteriaceae*, *Clostridium leptum*, and *Ruminococcus*	[200]
Cranberry extract polyphenols			Increased the abundance of *Akkermansia* species	[190]
Cranberry	Animal study (mice; high-fat diet)	200 mg/kg (berry extracts) for 9 weeks	Pyrosequencing (+) *Akkermansia*	[190]
Goji berry	Animal study (mice; colitis model)	1% (dry berry powder) for 10 weeks	*Bifidobacteria*, *Clostridium leptum*, *Fecalibacterium prazusnitzii*	[214]
Grape polyphenols			Increased the beneficial anaerobic gut bacteria, such as *Akkermansia muciniphila*	[186]
Grape polyphenols	Animal study (mice; high-fat diet)		Significantly reduced the *Firmicutes*/*Bacteroidetes* ratio Promoted the growth of *A. muciniphila*	[192]
Juçara	In vitro	1% (dry berry powder) for 24 h	Promoted abundance of *Bifidobacterium*, *Eubacterium rectale-Clostridium coccoide*, and *Bacteroides* spp.	[215]
Juçara	Animal study (rats; high-fat diet)	0.5 and 0.25% (dry berry powder) for 7 days	Promoted abundance of *Bifidobacterium*	[216]
Lingonberry	Animal study (mice; high-fat diet)	44% (dry berry powder) for 8 weeks	Promoted abundance of *Bacteroides*, *Parabacteroides*, *Clostridium* 106, (−) *Mucispirillum*, and *Oscillospira*	[195]
*Lonicera cerula* L.	Animal study (mice; high-fat diet)	1% (dry berry powder for 45 days)	Promoted abundance of *Bacteroides*, *Parabacteroides*, two genera from the order Bacteroidales (−) *Staphylococcus*, *Lactobacillus*, *Ruminococcus*, and *Oscillospira*	[191]
*Lycium ruthenicum*	In vitro (batch-culture fermentation)	1 g/L anthocyanins for 24 h	Promoted abundance of *Bifidobacterium*, *Allisonella* (−) *Prevotella*, *Dialister*, *Megamonas*, and *Clostridium*	[202]
*Plinia jaboticaba*	Animal study (healthy rats)	Juice for 2 and 7 weeks	Promoted abundance of *Lactobacillus*, *Bifidobacterium*, and *Enterobacteriaceae*	[217]
Procyanidin supplement			Significantly increased the β-diversity of intestinal microbiota and the *Bacteroidetes* abundance Reduced the *Firmicutes*/*Bacteroidetes* ratio Decreased the abundance of *Lachnospiraceae*	[218]
Resveratrol			Inhibited the growth of *Enterococcus faecalis*, and increased the growth of *Bifidobacterium* and *Lactobacillus*	[185]
Raspberry, cloudberry, and strawberry	In vitro	0.5–5 mg/mL of berries extracts (24–48 h)	Inhibited the growth of Gram-negative but not Gram-positive bacteria	[188]
*Schisandra chinensis*	Human study (obese volunteers)	6.7 g dry berry powder per day for 12 weeks	Promoted abundance of Bacteroides, *Akkermansia*, *Roseburia*, *Prevotella Bifidobacterium,* and (−) *Ruminococcus*	[199]
Sea buckthorn	In vitro	250 mg lyophilized fraction of small intestine digested berries for 72 h	Promoted abundance of *Bacteroides/Prevotella* and *Bifidobacteria*	[219]
		10% berry juice for 1 week (continuous gut model)	Promoted abundance of *Lactobacilli*, *Bacteroides*/*Prevotella*, and *Bifidobacteria*	[220]
Strawberry	Animal study (mice; diabetic model)	2.35% (dry berry powder) for 10 weeks	Promoted abundance of *Bifidobacterium*, (−) *Bacteroides*, and *Akkermansia*	[221]

## Data Availability

Not applicable.

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
