# Peer review of "Chemical Compounds of Berry-Derived Polyphenols and Their Effects on Gut Microbiota, Inflammation, and Cancer"

_molecules, 2022, doi:10.3390/molecules27103286_

Round 1

Reviewer 1 Report

  • The manuscript is clear, relevant for the field and presented in a well-structured manner, but the title does not fully reflect the content about the effects of the substances described on cancer cells. I propose to expand the title or divide the manuscript into two articles.
  • In chapter 4.1. Effects on gut microbiota, please standardize the style of writing the names of bacteria in italics.
  • The most of the articles cited are more than 5 years old, but this does not detract from their value.
  • The figures, tables and images are appropriate.
  • The conclusions are consistent with the evidence and arguments presented.

Author Response

Reviewer 1

  • The manuscript is clear, relevant for the field and presented in a well-structured manner, but the title does not fully reflect the content about the effects of the substances described on cancer cells. I propose to expand the title or divide the manuscript into two articles.

Response

The title was expanded and edited.

  • In chapter 4.1. Effects on gut microbiota, please standardize the style of writing the names of bacteria in italics.

Response

Checked and corrected.

  • The most of the articles cited are more than 5 years old, but this does not detract from their value.

Response

Recent references were added to improve the quality of the manuscript.

  • The figures, tables and images are appropriate.

Response

Thank you very much for your appreciation.

  • The conclusions are consistent with the evidence and arguments presented

Response

      Thank you very much for your appreciation.

Reviewer 2 Report

The paper is very interesting and well organized. In addition, it explores a relevant subject. The authors provided an extensively revision of the state of the art of berries polyphenols properties and applications. However, in order to make it a stronger paper, I suggest some points before it can be published:

  • A list of abbreviations would improve the manuscript
  • Due to the large size of the manuscript, I suggest an English revision to correct some typo mistakes
  • Line 99: please always write the full scientific name of the organism in the first time it appears in the text (V. vinifera).
  • Lines 90-135 and 136-164: too long paragraph. In general, there are too long paragraphs all over the manuscript. Please, smaller sentences will make it easier to read.
  • Item 2.1 focus mainly in grape fruits, why other berries are not explored? If there are low literature about it, this should be set in the text.
  • Lines 266-270: the effect of temperature could be more explored. Higher or lower temperatures are better to polyphenol production?
  • Table 1 is not mentioned or called in the manuscript text
  • Items 2.1 and 2.2 could be swapped to make the manuscript more fluid to read
  • Line 415: “anthocyanins (up to 5000 mg kg−1), which are responsible for the blue, purple, and red color as well as the high acceptance of these fruits” this sentence is repeated several times in manuscript, please correct it
  • There are some microorganisms scientific names that are not in italic in the text, please correct it.
  • Supplementary material figure 1 please correct adopted to adapted

Author Response

Reviewer 2

The paper is very interesting and well organized. In addition, it explores a relevant subject. The authors provided an extensively revision of the state of the art of berries polyphenols properties and applications. However, in order to make it a stronger paper, I suggest some points before it can be published:

  • A list of abbreviations would improve the manuscript

Response

A list of abbreviations was added.

  • Due to the large size of the manuscript, I suggest an English revision to correct some typo mistakes

Response

The manuscript was checked and revised by a native speaker English.

  • Line 99: please always write the full scientific name of the organism in the first time it appears in the text (V. vinifera).

Response

Checked and corrected.

  • Lines 90-135 and 136-164: too long paragraph. In general, there are too long paragraphs all over the manuscript. Please, smaller sentences will make it easier to read.

Response

These paragraphs have been checked and revised.

  • Item 2.1 focus mainly in grape fruits, why other berries are not explored? If there are low literature about it, this should be set in the text.

Response

Thank you for this comment.

Indeed, this remark was raised by another reviewer.

This species is one of other species that can produce berries. The evaluation of the regulation of polyphenol biosynthesis was investigated using this plant as a "study model". This is why we used this plant to explain the mode of polyphenol synthesis. On the other hand, studies concerning the other species have not approached this regulation of synthesis in a very detailed way.

  • Lines 266-270: the effect of temperature could be more explored. Higher or lower temperatures are better to polyphenol production ?

Response

Checked and improved.

Table 1 is not mentioned or called in the manuscript text

Response

Checked and corrected.

  • Items 2.1 and 2.2 could be swapped to make the manuscript more fluid to read

Response

Checked and corrected.

  • Line 415: “anthocyanins (up to 5000 mg kg−1), which are responsible for the blue, purple, and red color as well as the high acceptance of these fruits” this sentence is repeated several times in manuscript, please correct it

Response

Checked and corrected.

  • There are some microorganisms scientific names that are not in italic in the text, please correct it.

Response

Checked and corrected.

  • Supplementary material figure 1 please correct adopted to adapted

Response

Checked and corrected.

Reviewer 3 Report

Title: Beneficial effects of berries-derived polyphenols on gut microbiota

Authors: Abdelhakim Bouyahya, Nasreddine El Omari, Naoufal EL Hachlafi,

Meryem El Jemly, Mryam Hakkour, Abdelaali Balahbib, Naoual El Menyiy, Saad Bakrim, Hanae Naceiri Mrabti, Aya Khouchlaa, Mohamad Fawzi Mahomoodally, Michelina Catauro, Domenico Montesano, Gokhan Zengin

Comments:

This review will provide an overview of the major bioactive compounds in berry fruits and their potential modulation of the gut microbiota and gut health, as well as their health-promoting properties in relation to metabolic disorders and various cancers. In addition, data from studies on the bioavailability, metabolism, and mechanisms of polyphenols from various berry fruits will be presented, as well as technological approaches to improve the medicinal value of these compounds.

Important major points are listed below:

1: The abstract should be rewritten as it contains many repetitions and appears unstructured.

2: Expression Chapter 2 "Generality about" sounds strange in this context, when rather "general information about...".

3: Chapter 2.1. "Biosynthesis and Regulation" is about V. vinifera (see pages 2-6), but according to the title it should be about berries in general (other species are mainly listed in Table 1, but not included in the main text); moreover, biosynthesis and regulating factors are described in a very unstructured/confusing way

4: Table 1 is not integrated into the text!

5: Chapter 2.2. "Chemical composition" is very short in contrast to the previous chapter. Here, a subdivision into the different components with explanation of their properties would be nice

6: Page 7, top line: Please check spelling (Wildberryand...).

7: Page 9, top line: Please check spelling (floribundumKunth) and further down (vinifera)samples).

8: Chapter headings should be clarified, e.g., 3.2.-3.4., 4.2., 4.2.1., 4.2.2., 5.1. (What kind of effect?), and 4.1. (Where is the difference from heading 4.?).

9: Titles 3.2., 3.3. and 3.4. with "Effects of..." very unspecific, which effects are concerned?

10: Also, both chapter 3.2. and 3.3. and 3.4. again discuss "bioavailability" in great detail, although this was previously discussed in detail in 3.1.

11: I miss throughout chapter 3 the pharmacokinetic processes/mechanisms/effects described in the title.

12: The title of chapter 4 and chapter 4.1. says more or less the same thing. Where is the difference?

13: Chapter 4, which deals with the microbiota (and which, according to the title, should actually be the focus), is very short compared to the previous chapters. Only a few berry species are described in the subchapters.

14: The subdivision into "gut microbiota" (Chapter 4.1.) and "colon microbiota" (Chapter 4.2.) makes only limited sense; some statements with the same references are repeated in similar wording, e.g. page 23 lines 683-687 and page 18 lines 622-627. The further subdivision into "pathogenic" and "beneficial" bacteria also makes little sense here, since these have already been dealt with in Chapters 4.1. and 4.2. It would be better to integrate these chapters into the preceding chapters or preceding chapters into these chapters, so that correlations become clearer and a plausible conclusion can be drawn from them. Furthermore, the question arises as to why further subdivisions were made for "colon microbiota" but not for "gut microbiota"?

15: Chapter 5.1.2: Why are in vitro and human studies covered in one chapter? Clinical studies in humans also belong to in vivo studies (see Chapter 5.1.1.).

16: The naming of the individual chapters is very inconsistent: e.g., "5.1. effects on intestinal inflammation" and "5.2. positive effects on various cancers by microbiota" (the same applies to 5.3.): stimulation: Chapter 5.1. e.g., "effects on gastrointestinal diseases" (because inflammation is also the basis of cancer) and in Chapter 5.2. "effects on different types of cancer", so that there is more structure and clarity for the reader.

17: In chapter 5.2.1. "Colorectal cancer" there is again a subdivision into "in vivo studies" and "in vitro studies", however, in contrast to the previous chapter, no separate numbering is used here; moreover, there is then a subdivision into "clinical studies" (previously, clinical studies were treated as if they had been described together with in vitro studies; the same is then repeated again in chapter 5.2.2.) inconsistent and illogical!

18: Figure 4: for a picture that contains so many metabolic pathways/mechanisms, I miss a descriptive legend (there is also an error in the title of the legend that makes the content somewhat incomprehensible)

19: Figure 5: is it specifically about strawberries (as shown in the picture) or about "berry polyphenols" (as described in the legend)?

20: In chapter 5.2.3. and chapter 5.2.4. there is suddenly no subdivision at all between "in vivo" and "in vitro" studies

21: Page 13, line 374: Please check the reference! Called Bermundez-Soto, citing Burgos-Edwards.

22: Page 15, line 483: It should be Mueller instead of Muller (reference 108).

23: Page 16, line 516: It should be Van de Velde instead of Velde (reference 139).

24: Line 28, In vitro studies on CRC: It should be mentioned here that resveratrol is anticarcinogenic in many ways and even has chemosensitizing effects, e.g.: doi: 10.1016/j.bcp.2015.08.105.

doi: 10.3390/nu10070888.

25: Figure 4: A nice image that should be made larger for readability. Also, the discrepancy that the text refers to the figure for proteins and conditions that are not mentioned there (e.g., TME, p53) should be revised.

26: Section 5.2.2.-5.2.4: Where is there a specific reference to the "gut microbiota" in these cancers? Please clearly explain the references in this regard or delete the chapters and write about the related and addressed metabolic diseases instead.

28: The discussion on the topic (chapter 6) comes a bit too short in view of the much information in the previous chapters

29: A complete list of abbreviations is necessary.

30: English in general needs improvement. All text needs to be revised by a native English speaker.

Formatting

- Author address numbering needs to be corrected.

- NF-kB in line 44.

- Formatting adjustments in tables, e.g., page 21.

- Insert period at end of sentence: Page 28, line 951.

- Delete comma: page 34, line 1188.    

Author Response

Reviewer 3

This review will provide an overview of the major bioactive compounds in berry fruits and their potential modulation of the gut microbiota and gut health, as well as their health-promoting properties in relation to metabolic disorders and various cancers. In addition, data from studies on the bioavailability, metabolism, and mechanisms of polyphenols from various berry fruits will be presented, as well as technological approaches to improve the medicinal value of these compounds.

Important major points are listed below

1: The abstrac3t should be rewritten as it contains many repetitions and appears unstructured.

Response

The abstract was checked and improved.

2: Expression Chapter 2 "Generality about" sounds strange in this context, when rather "general information about...".

Response

Checked and corrected.

3: Chapter 2.1. "Biosynthesis and Regulation" is about V. vinifera (see pages 2-6), but according to the title it should be about berries in general (other species are mainly listed in Table 1, but not included in the main text); moreover, biosynthesis and regulating factors are described in a very unstructured/confusing way

Response

Thank you for this comment.

Indeed, this species is one of other species that can produce berries. The evaluation of the regulation of polyphenol biosynthesis was investigated using this plant as a "study model". This is why we used this plant to explain the mode of polyphenol synthesis. On the other hand, studies concerning the other species have not approached this regulation of synthesis in a very detailed way.

4: Table 1 is not integrated into the text!

Response

Checked and corrected.

5: Chapter 2.2. "Chemical composition" is very short in contrast to the previous chapter. Here, a subdivision into the different components with explanation of their properties would be nice

Response

This part was revised and improved.

6: Page 7, top line: Please check spelling (Wildberryand...).

Response

Checked and corrected.

7: Page 9, top line: Please check spelling (floribundumKunth) and further down (vinifera)samples).

Response

Checked and corrected.

8: Chapter headings should be clarified, e.g., 3.2.-3.4., 4.2., 4.2.1., 4.2.2., 5.1. (What kind of effect?), and 4.1. (Where is the difference from heading 4.?).

Response

Checked and titles were changed. Some titles were ordered and changed.

9: Titles 3.2., 3.3. and 3.4. with "Effects of..." very unspecific, which effects are concerned?

Response

They were checked and revised.

10: Also, both chapter 3.2. and 3.3. and 3.4. again discuss "bioavailability" in great detail, although this was previously discussed in detail in 3.1.

Response

Titles were checked and revised.

In 3.1. we discussed in general the bioavailability of bioactive compounds their metabolism but in 3.2. and 3.3. the specificity of the subgroups (phenolic acids, flavonoids, etc.) were highlighted.

11: I miss throughout chapter 3 the pharmacokinetic processes/mechanisms/effects described in the title.    

Response

The titles were changed. In this section we highlight the bioavailability and the metabolism.  However, details concerning the mechanisms of pharmacokinetic ways have not been descried in the literature.    

12: The title of chapter 4 and chapter 4.1. says more or less the same thing. Where is the difference?

Response

Titles were checked and revised importantly.

13: Chapter 4, which deals with the microbiota (and which, according to the title, should actually be the focus), is very short compared to the previous chapters. Only a few berry species are described in the subchapters.

Response

Because the data related to these aspects have not been found.

14: The subdivision into "gut microbiota" (Chapter 4.1.) and "colon microbiota" (Chapter 4.2.) makes only limited sense; some statements with the same references are repeated in similar wording, e.g. page 23 lines 683-687 and page 18 lines 622-627. The further subdivision into "pathogenic" and "beneficial" bacteria also makes little sense here, since these have already been dealt with in Chapters 4.1. and 4.2. It would be better to integrate these chapters into the preceding chapters or preceding chapters into these chapters, so that correlations become clearer and a plausible conclusion can be drawn from them. Furthermore, the question arises as to why further subdivisions were made for "colon microbiota" but not for "gut microbiota"?

Response

This section was checked and its parts were re-subdivided.

15: Chapter 5.1.2: Why are in vitro and human studies covered in one chapter? Clinical studies in humans also belong to in vivo studies (see Chapter 5.1.1.).

Response

Thank you for your pertinent comment, we have proceeded with the required modifications with separate numbering as follows:

Chapter 5.1.1

5.1.1.1.In vitro studies

5.1.1.2.In vitro studies

5.1.1.3.Clinical trials

Chapter 5.1.2

5.1.2.1.In vitro studies

5.1.2.2.In vitro studies

5.1.2.3.Clinical trials

16: The naming of the individual chapters is very inconsistent: e.g., "5.1. effects on intestinal inflammation" and "5.2. positive effects on various cancers by microbiota" (the same applies to 5.3.): stimulation: Chapter 5.1. e.g., "effects on gastrointestinal diseases" (because inflammation is also the basis of cancer) and in Chapter 5.2. "effects on different types of cancer", so that there is more structure and clarity for the reader.

Response

-Concerning the re-naming of chapter 5.1. effects on intestinal inflammation, we suggest not to change this title for the following purposes:

Inflammation is one of the risk factors for cancer, especially cancers related to infections (in fact, chronic inflammation related to the intestinal microbiota is only one of several events that predispose to tumor transformation, therefore, intestinal inflammation does not necessarily mean the occurrence of cancer, often it remains in the form of ulcers.

-Chapter 5.2. "effects on different types of cancer: Checked and title was changed

17: In chapter 5.2.1. "Colorectal cancer" there is again a subdivision into "in vivo studies" and "in vitro studies", however, in contrast to the previous chapter, no separate numbering is used here; moreover, there is then a subdivision into "clinical studies" (previously, clinical studies were treated as if they had been described together with in vitro studies; the same is then repeated again in chapter 5.2.2.) inconsistent and illogical!

Response

We have followed the same logic and subdivision for chapters 5.2.1 and 5.2.2 (in vitro studies, in vivo studies, and clinical trials) by using separate numbering

18: Figure 4: for a picture that contains so many metabolic pathways/mechanisms, I miss a descriptive legend (there is also an error in the title of the legend that makes the content somewhat incomprehensible)

Response

Checked, revised and the legend was added.

19: Figure 5: is it specifically about strawberries (as shown in the picture) or about "berry polyphenols" (as described in the legend)?

Response

Checked and revised

20: In chapter 5.2.3. and chapter 5.2.4. there is suddenly no subdivision at all between "in vivo" and "in vitro" studies

Response

Checked and corrected.

21: Page 13, line 374: Please check the reference! Called Bermundez-Soto, citing Burgos-Edwards.

Response

Checked and corrected.

22: Page 15, line 483: It should be Mueller instead of Muller (reference 108).

Response

Checked and corrected.

23: Page 16, line 516: It should be Van de Velde instead of Velde (reference 139).

Response

Checked and corrected.

24: Line 28, In vitro studies on CRC: It should be mentioned here that resveratrol is anticarcinogenic in many ways and even has chemosensitizing effects, e.g.: doi: 10.1016/j.bcp.2015.08.105.

doi: 10.3390/nu10070888.

Response

Checked and corrected

25: Figure 4: A nice image that should be made larger for readability. Also, the discrepancy that the text refers to the figure for proteins and conditions that are not mentioned there (e.g., TME, p53) should be revised.

Response

The figure and its caption were checked and revised

26: Section 5.2.2.-5.2.4: Where is there a specific reference to the "gut microbiota" in these cancers? Please clearly explain the references in this regard or delete the chapters and write about the related and addressed metabolic diseases instead.

Response

These parts were extensively revised and improved by adding news references

28: The discussion on the topic (chapter 6) comes a bit too short in view of the much information in the previous chapters

Response

The chapter 6 describes only a conclusion and general perspectives.

29: A complete list of abbreviations is necessary.

Response

A list of abbreviations was added

30: English in general needs improvement. All text needs to be revised by a native English speaker.

Response

The language was checked and corrected

Formatting

- Author address numbering needs to be corrected.

Response

Checked and revised.

- NF-kB in line 44.

Response

Checked and corrected.

- Formatting adjustments in tables, e.g., page 21.

Response

Tables were formatted.

- Insert period at end of sentence: Page 28, line 951.

Response

Checked and corrected.

- Delete comma: page 34, line 1188.    

Response

Checked and corrected.

Round 2

Reviewer 1 Report

The authors have made satisfactory changes and corrections.

Reviewer 3 Report

The authors have satisfactorily addressed the concerns raised in the original version. The revised version is significantly improved. No further concerns.